# A mechanical ratchet drives unilateral cytokinesis

Alison Kickuth[1,2,3], Urša Uršič[1,2,3], Michael F. Staddon[1,2,3,4] & Jan Brugués[1,2,3 ✉]

The canonical mechanism that drives cell division comprises the formation and constriction of a contractile actin ring[1–3]. However, this mechanism is not compatible with the early development of many vertebrates[4–9]. Yolk-anchored embryos typically cannot form a complete ring during early cleavage divisions, but it remains unclear how a partial circular arc with loose ends can divide the cell. Here, by combining laser ablation of the cytokinetic band with rheological measurements in vivo, we show that stiffening of the bulk cytoplasm, mediated by the interphase microtubule network, stabilizes the contractile band by anchoring it along its length during growth. Conversely, as the cell cycle progresses, the cytoplasm fluidizes, diminishing band–cytoplasmic anchoring and facilitating band ingression. This dynamic interplay between stability and growth versus instability and ingression repeats for several cell cycles until division is complete, resulting in a mechanical ratchet that drives cell division. Our study underscores the role of temporal control over cytoplasmic rheology as a key feature that drives unilateral cytokinesis in the absence of a closed actin ring.

In the canonical model of cell division, a contractile actomyosin ring assembles around the cell equator and symmetrically contracts to constrict the cell[1,10–13]. However, in many species the cleavage furrow ingresses unilaterally during early development[5–9], either holoblastically (complete division with an initially incomplete ring) or meroblastically (partial division with undivided yolk). Examples of species in which cell division occurs without a complete ring include cephalopods, elasmobranchs (sharks and rays), teleosts, birds and reptiles[5,6], as well as the mammal platypus[7–9], making this a widely used yet unexplored process. Notably, these embryonic cells divide using a contractile band that surrounds only part of the cell and has loose ends. It is unclear how this incomplete ring is stabilized and can generate force to divide the cell, as contraction of an open-ended band would intuitively cause it to collapse under tension. Here we study the well-established model organism zebrafish (*Danio rerio*), whose large blastomeres (diameter approximately 700 μm) undergo unilateral cytokinesis during early developmental stages[14].

In early zebrafish cell division, after the DNA has been divided by the spindle in mitotic phase (M-phase), large microtubule asters[15–17] start growing out to span the entire cytoplasm in interphase. These interphase asters determine the cleavage plane[18,19], where the contractile band starts to form. The band starts growing in the centre and then elongates on both sides. However, the band never fully surrounds the embryonic cell, but instead stops growing when it reaches the yolk. In this Article, we use interphase to refer to the phase during which microtubule asters span the cytoplasm, and use M-phase to refer to the mitotic phase in which the microtubule asters have been disassembled and spindles are formed.

In this study, we combine femtosecond laser ablation of the contractile band with rheological measurements of the cytoplasm to investigate the physical mechanisms that underlie unilateral cytokinesis. We find that microtubule-mediated stiffening of the cytoplasm anchors the band along its length, and fluidization of the cytoplasm in the next cell cycle allows its ingression. We propose a trade-off between stability and growth versus instability and ingression of the band, which establish cell division by alternating in a ratchet-like manner.

## Anchoring of the contractile actin band

To investigate the mechanisms of how an open-ended actin band can be set up and ingress, we imaged its formation during the first cell division in zebrafish embryo development. As previously reported, the actin band forms where astral microtubules meet to form the cleavage plane[18,20–22]. The forming band is surrounded by an actin-depleted zone as it continues to grow along the microtubule overlap zone until it reaches the cytoplasm-yolk boundary (Fig. 1a,b, Supplementary Fig. 1c and Supplementary Video 1 for top view, Fig. 1c for schematic, and Supplementary Fig. 1a,b for side view). One possibility to set up an open-ended actin band is that its initial formation does not involve active contractility[23]. This possibility would explain why the band does not collapse during its formation. To test whether the initial formation of the band involves active contractility, we used live confocal imaging in combination with laser ablation during band formation. Cutting the band resulted in recoil of 7.80 ± 0.02 μm in 10 s (Supplementary Fig. 1g). These experiments showed that the actin band is contractile as it is forming, raising the question of how it is stabilized. Notably, the recoil stopped after 10 s, which could be due to several reasons: either the tension of the band relaxed, the actin band rapidly healed and the severed ends reconnected, or the band was locally anchored, which prevented further recoil. To investigate these possibilities, we cut the band every 10–20 s, ensuring that it remained interrupted (Fig. 1d), thereby preventing it from healing. Notably, after 2 min of continuous band disconnection, the remaining band on either side of

[1]Cluster of Excellence Physics of Life, Technische Universität Dresden, Dresden, Germany. [2]Max Planck Institute of Molecular Cell Biology and Genetics, Dresden, Germany. [3]Max Planck Institute for the Physics of Complex Systems, Dresden, Germany. [4]Center for Systems Biology Dresden, Dresden, Germany. ✉e-mail: jan.brugues@tu-dresden.de

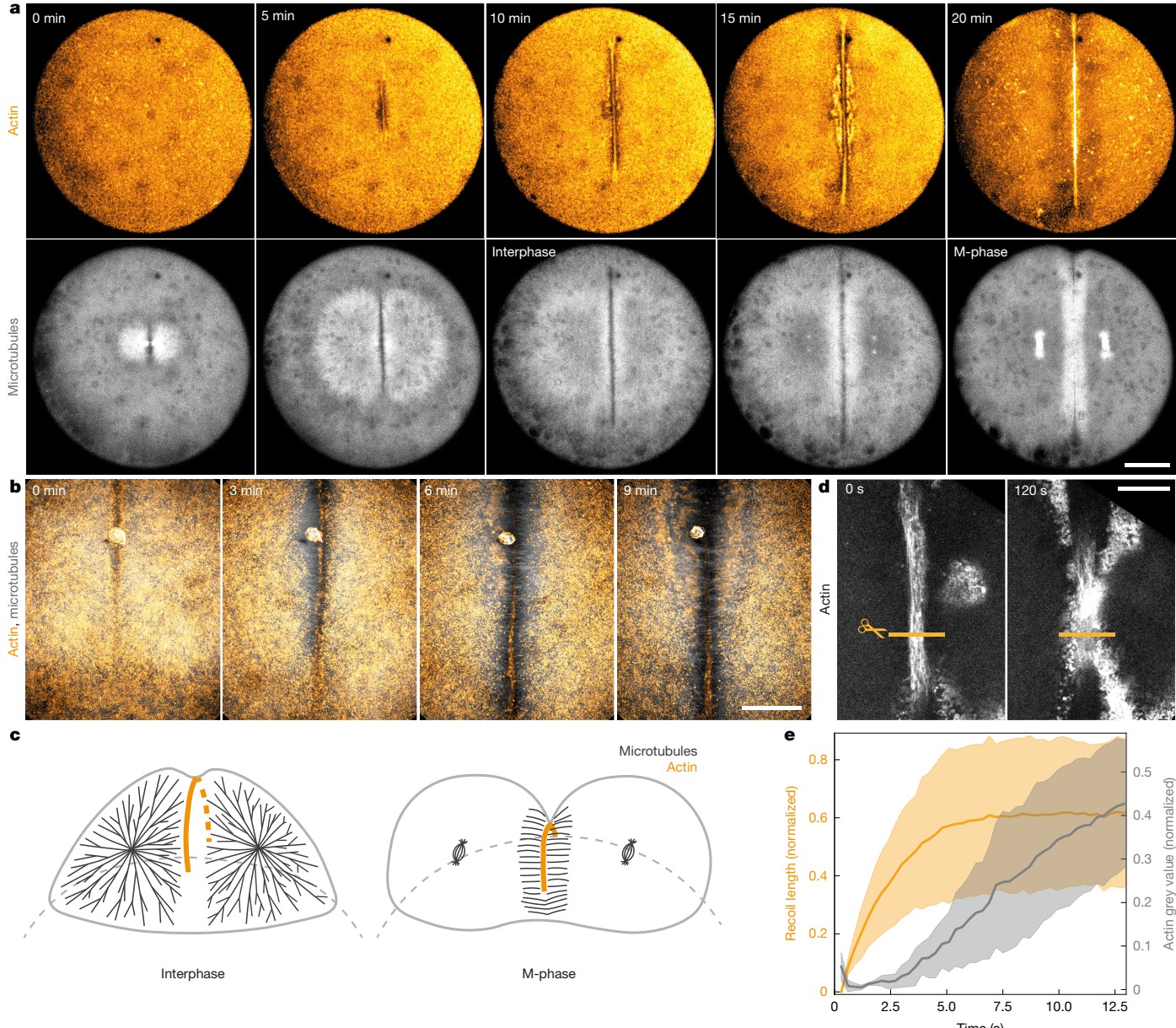

**Fig. 1 | Laser ablation reveals local anchoring of the contractile band.**
**a**, Zebrafish (*D. rerio*) embryos during the first cell division. Maximum intensity projection of actin (labelled by utrophin–mCherry) and microtubules (labelled by DCX–GFP). The actin band formation is visible from 5 min onwards and is characterized by an actin band, surrounded by an actin-depleted zone. Scale bar, 100 μm. **b**, Maximum intensity projection of embryo labelled for actin and microtubules visualizing the band formation at high resolution. Scale bar, 50 μm. **c**, Schematic of the zebrafish embryo during the first to second cell cycle, visualizing microtubule morphology in interphase and M-phase. **d**, Single confocal imaging plane of the contractile band during continuous laser cuts for 2 min (left, before cuts; right, after continuous band interruption). Actin is shown in grey and the cut area is indicated by a line. Scale bar, 20 μm. **e**, Length of the band-ablation recoil and the actin grey value between the cut ends (representing the healing of the band) in *n* = 4 embryos. Solid lines show the mean, shaded regions show s.d.

the cut was still able to ingress, despite the interruption (Fig. 1d, right and Supplementary Fig. 1d), whereas the cut area did not ingress. By contrast, in an unperturbed embryo, the band ingresses homogeneously (Supplementary Fig. 1e). Furthermore, although the healing of the band was very fast (approximately 15 s), our data suggest that the recoil stopped before the band fully healed from a single cut (Fig. 1e and Supplementary Fig. 1h, grey value recovered). The band also stopped retracting when multiple subsequent cuts were made in the same position (Supplementary Fig. 2a), indicating a different or additional mechanism that halts the recoil, such as local anchoring. Therefore, our data suggest that the active contractile band is set up by locally anchoring it throughout its length.

## Microtubules anchor the contractile band

Next, we sought to investigate the potential components that could act as the anchor for the contractile band. As main candidates we investigated actin and microtubules, as they are mechanical elements of the cell[24–26] and are involved in the formation of the actin band[22,27]. Although the actin cortex encloses the band, the cortex seems to be disconnected from it and appears to 'dissolve' in the area around band formation, leading to a gap between the two structures (Figs. 1a,b, 2f,g and 3a and Supplementary Figs. 1b and 2d). Laser cuts of the actin cortex (away from the band) did not open up, suggesting that the cortex is under low tension (Supplementary Fig. 2c and Supplementary Video 2).

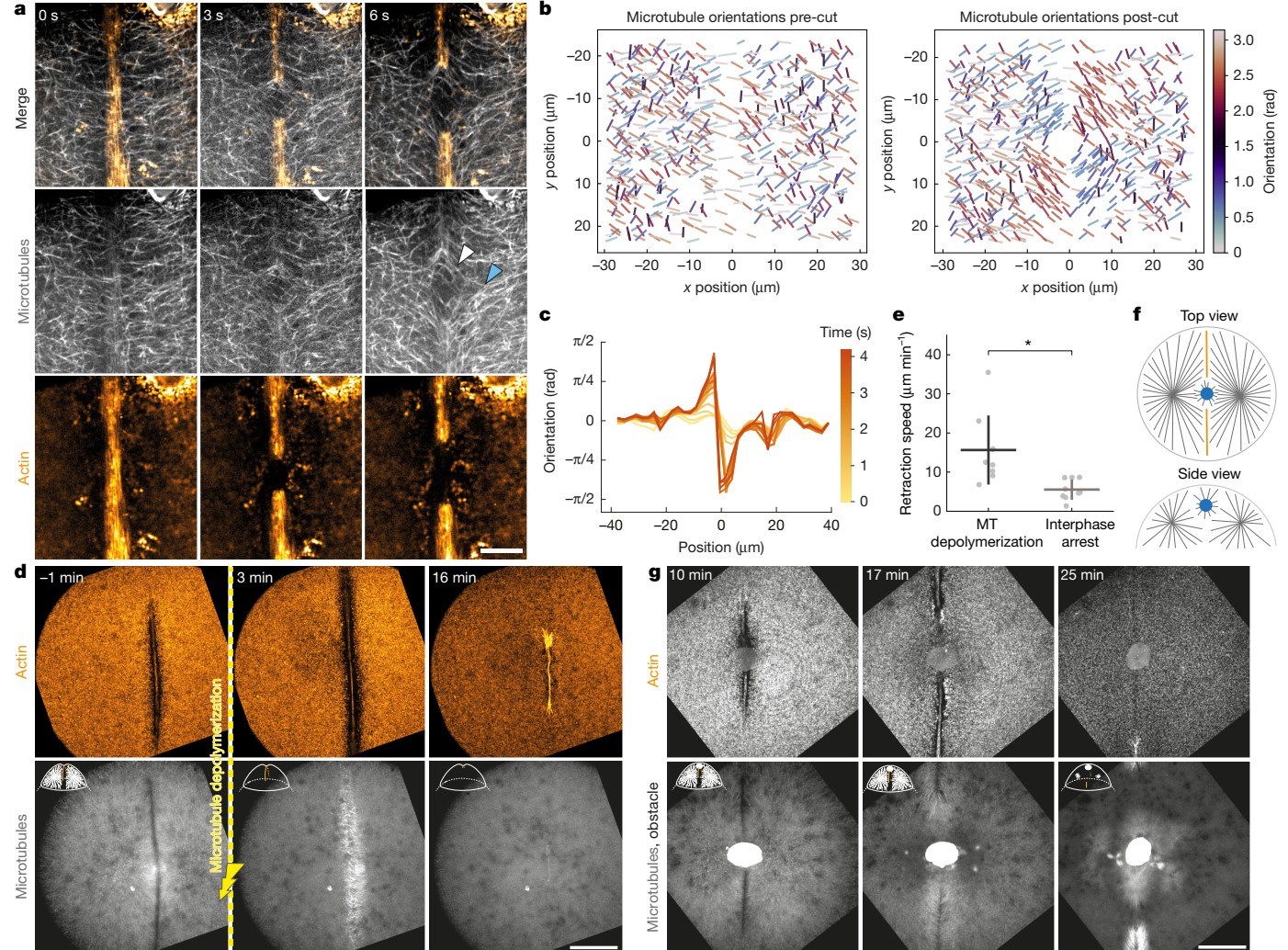

**Fig. 2 | The contractile band is anchored to microtubules. a**, Laser cut on the contractile band. The white arrowhead indicates displacement of microtubules with the actin band recoil; the blue arrowhead indicates the other side of microtubules that is not displaced. Scale bar, 20 μm. **b**, Microtubule orientations before and approximately 6 s after laser ablation of the band, from $n = 7$ embryos (more than 100 measurements per embryo per time step). **c**, Automatically detected microtubule orientation profile adjacent to the contractile band, during laser ablation of the band from $n = 1$ embryo. **d**, Embryo (top view) treated with the photoactivatable microtubule inhibitor SbTubA4P. Yellow line indicates photoactivation. Scale bar, 100 μm. **e**, Quantification of band retraction velocity of microtubule-depolymerized embryos ($15.6 \pm 8.8$ μm min$^{-1}$ (mean ± s.d.)) and interphase arrested embryos ($5.5 \pm 2.6$ μm min$^{-1}$) measured using particle image velocimetry (PIV). $n = 9$ embryos per condition; $P = 0.00457$, two-sided Student's $t$-test. MT, microtubule. **f**, Schematic representation of how the obstacle interrupts the contractile band. **g**, Embryo (top view) during first cell division, the contractile band is interrupted by an obstacle in the cytoplasm (visible in actin and microtubule channel). Scale bar, 100 μm.

As a control, when tension was induced by triggering wound-healing response in the cortex, subsequent cuts of the cortex did recoil (Supplementary Fig. 2f and Supplementary Video 2), confirming that the cortex is able to recoil when tension is induced via wound-healing. Finally, laser cuts of the band and the adjacent cortex did not consistently lead to recoil of the cut cortex with the band (Supplementary Fig. 2d and Supplementary Video 2). Together, these experiments suggest that the actin cortex itself is not anchoring the band.

Microtubules, however, are present at the cleavage furrow and throughout the cytoplasm. In interphase, when the band starts to form, the microtubules form large structures called asters[16,18,28] that define the cleavage plane[18]. In addition, they are also a mechanical element of the cell and are able to exert forces[29], which could help to stabilize the band. Furthermore, our laser ablation experiments showed that the microtubules next to the band consistently bend and splay open when the band is cut, indicating a mechanical connection between microtubules and the contractile band (Fig. 2a–c, Supplementary Fig. 2b and Supplementary Video 3). Of note, whereas the microtubule ends close to the band moved when the band recoiled, the ends of the microtubules further from the cut were not affected by the recoil, indicating that they might be physically held in place (Fig. 2a). To further test the mechanical role of microtubules in anchoring the actin band, we disrupted microtubule polymerization by using the photoactivatable microtubule inhibitor SbTubA4P[30], which enabled us to perturb microtubules at specific times during band formation. Upon photoactivation, SbTubA4P sequesters the free tubulin from the cytoplasm, thereby preventing further microtubule growth and leading to their rapid disappearance due to microtubule turnover. We activated the inhibitor 5 min after the onset of contractile band formation, leading to a rapid and complete depolymerization of microtubule asters after 3 min (Fig. 2d and Supplementary Video 4). Simultaneously to this depolymerization, the contractile band rapidly retracted (with a velocity of $15.6 \pm 8.8$ μm min$^{-1}$; Fig. 2e) and collapsed, which led to a failure of cytokinesis and left the cell undivided. Taken together, these

experiments indicate that microtubules are mechanically connected to the band and are essential for its stability, whereas the membrane or actin cortex were not sufficient to anchor the band when microtubules were absent. Therefore, we conclude that microtubules are a crucial anchoring element in setting up the actin band.

## Interphase microtubules stabilize the band

Our data suggest that microtubules are a crucial anchoring element in establishing the actin band. However, microtubules also have a signalling role to form and maintain the band[22,27]. To disentangle these two roles, we sought to find a perturbation that would locally perturb microtubules without affecting their signalling role elsewhere. To this end, we introduced a physical obstacle into the embryo. The obstacle consisted of an oil droplet that we injected into the centre of the cytoplasm. The obstacle causes a local steric hindrance to band formation, and the surface coating of the droplet nucleated microtubules, thereby creating an area in which the microtubule midzone was perturbed—in the proximity of the obstacle—whereas the microtubules in the rest of the cell were unaffected (Fig. 2f,g and Supplementary Video 5). Owing to the local interruption of signalling, the actin band was interrupted where the obstacle was, leading to two shorter and disconnected bands forming on either side. Remarkably, during interphase, when the microtubule asters filled the cytoplasm, the two half-bands were stable and grew. However, as the cell cycle progressed and the interphase microtubule asters disassembled, the bands concomitantly collapsed. This experiment shows that beyond signalling, microtubules contribute mechanically to stabilize the actin band.

The observation that retraction of the contractile band only happened when microtubules were not present in the bulk cytoplasm led us to hypothesize that the bulk microtubule network formed by the interphase asters might influence the mechanics of the cytoplasm, stiffening the cytoplasm to provide a rigid scaffold that stabilizes the band. We predicted that arresting the cell cycle in interphase, and thereby maintaining the microtubule asters, should stabilize the band indefinitely. We used cycloheximide, an inhibitor of translation, to arrest the cell cycle in interphase. In this condition, the formation of the band proceeded similarly to the unperturbed embryo. Twenty minutes after the beginning of interphase (10 min beyond regular interphase duration), the band started to slowly retract (Fig. 3a, Supplementary Fig. 3c,d and Supplementary Video 6). However, unlike the microtubule-depolymerized case, this appeared to be an overall contraction of the band and cytoplasm rather than a collapse: the band remained stable and attached all the way until the yolk–cytoplasm boundary and the whole cytoplasm was moved as the band shortened, indicating a slight ingression at the band ends. In addition, the retraction seen in interphase is significantly slower ($5.5 \pm 2.6 \ \mu m \ min^{-1}$) than the retraction of the actin band when microtubules were depolymerized ($15.6 \pm 8.8 \ \mu m \ min^{-1}$) (Fig. 2e). Finally, we directly stabilized microtubules by treating the embryo with taxol[31] (Supplementary Fig. 3a,b and Supplementary Video 7). The taxol treatment severely disrupted the physiology of the microtubule network, but at sufficiently low taxol concentrations (Supplementary Fig. 3a) the band was able to form and remained in place over multiple cell cycles. Together, the band remained stable wherever microtubule asters were present.

If the interphase-arrested band is indeed mechanically supported by the cytoplasm stiffening through the microtubule asters, we expected this stability to be immediately lost upon microtubule depolymerization. To test this prediction, we combined cycloheximide treatment and injection of the photoactivatable microtubule inhibitor SbTubA4P. This combination caused arrest of the cell cycle in interphase, with microtubule asters, but allowed their depolymerization upon light exposure. We allowed the band to contract and retract, while cytoplasm ingressed at the band ends. At this stage, we depolymerized microtubules, releasing the embryo from the stabilization of the microtubule asters. This

microtubule depolymerization caused the cytoplasm to rapidly flow back to its original shape (white arrow in Fig. 3b and Supplementary Video 8), while the band promptly collapsed. Together, these results suggest that stiffening of the cytoplasm by bulk microtubule asters mechanically stabilize the band.

## Cytoplasm stiffens in interphase

The cytoplasm rheology is known to be changed by the cytoskeleton throughout the cell cycle in other systems[20,32–34]. To investigate whether bulk microtubules influence cellular mechanics in the zebrafish embryo, we quantified the material properties of the cytoplasm throughout the first cell cycles using magnetic and optical tweezers and ferrofluid droplets.

For the magnetic tweezer experiments, we injected fluorescent magnetic beads into the zebrafish embryo cytoplasm at the 1-cell stage and imaged the embryo throughout the first division. During this time, we applied repeated force pulses to the beads using magnetic tweezers (Fig. 3c,f). We applied the magnetic force to the beads for 5 s, followed by a 15-s pause before the next pulse. We inferred the viscoelastic properties of the cytoplasm from the bead force–displacement curves, while keeping track of the cell cycle through microtubule network morphology. The displacement of the beads over the course of a pulse was significantly lower in interphase as compared to M-phase (Fig. 3d,g), indicating stiffening of the cytoplasm in interphase. Consistently, the fraction of recovery to the initial position 15 s after the pulse was significantly higher in interphase (Fig. 3e). In summary, magnetic tweezer experiments are consistent with interphase microtubules stiffening the cytoplasm.

However, as detailed analysis of the magnetic tweezers data is typically limited to the model it is fitted to, we used optical tweezers to directly measure cytoplasm rheology in terms of a complex shear modulus ($G^*$; refs. 32,35). We injected polystyrene beads into the cytoplasm and measured the rheology across frequencies from 0.5 Hz to 2,056 Hz. We found that for high frequencies, the viscous modulus is dominant over the elastic modulus. For low frequencies, the elastic modulus becomes comparable and slightly dominates over the viscous one in interphase, indicating that the cytoplasm is more solid (Fig. 3h and Supplementary Fig. 4a), consistent with previous measurements in other systems[32]. Of note, the ratio of elastic and viscous moduli between interphase and M-phase are roughly constant (around threefold larger in interphase) across all measured frequencies (Supplementary Fig. 4b) and is consistent with the stiffening observed with magnetic tweezers. To further compare these two measurements, we converted the magnetic tweezer creep response[35] to $G^*$ (Fig. 3h). This procedure showed that these two measurements are remarkably consistent with each other in magnitude and trend, and reveal a threefold increase in viscosity and elasticity of the cytoplasm from M-phase to interphase, corresponding to presence or absence of microtubule asters.

To test whether the observed changes in material properties were caused by the microtubule asters, we repeated the magnetic tweezers measurements in embryos treated with nocodazole and taxol, in which microtubules were depolymerized or stabilized, respectively. In these conditions, we found that beads in embryos treated with taxol behaved similarly to beads in interphase. In turn, beads in embryos treated with nocodazole exhibited significantly greater displacement compared with those in taxol-treated embryos during both interphase and M-phase (Fig. 3d).

In line with the magnetic and optical tweezers experiments, ferrofluid droplets[36] injected into the cytoplasm and deformed under a homogeneous magnetic field behaved consistently with the stiffening of the cytoplasm in interphase. Under constant magnetic field, the deformation of the ferrofluid droplet became faster in M-phase, when the microtubules around the droplet disassembled. Correspondingly, when the deformation of the droplet was started in M-phase, the deformation

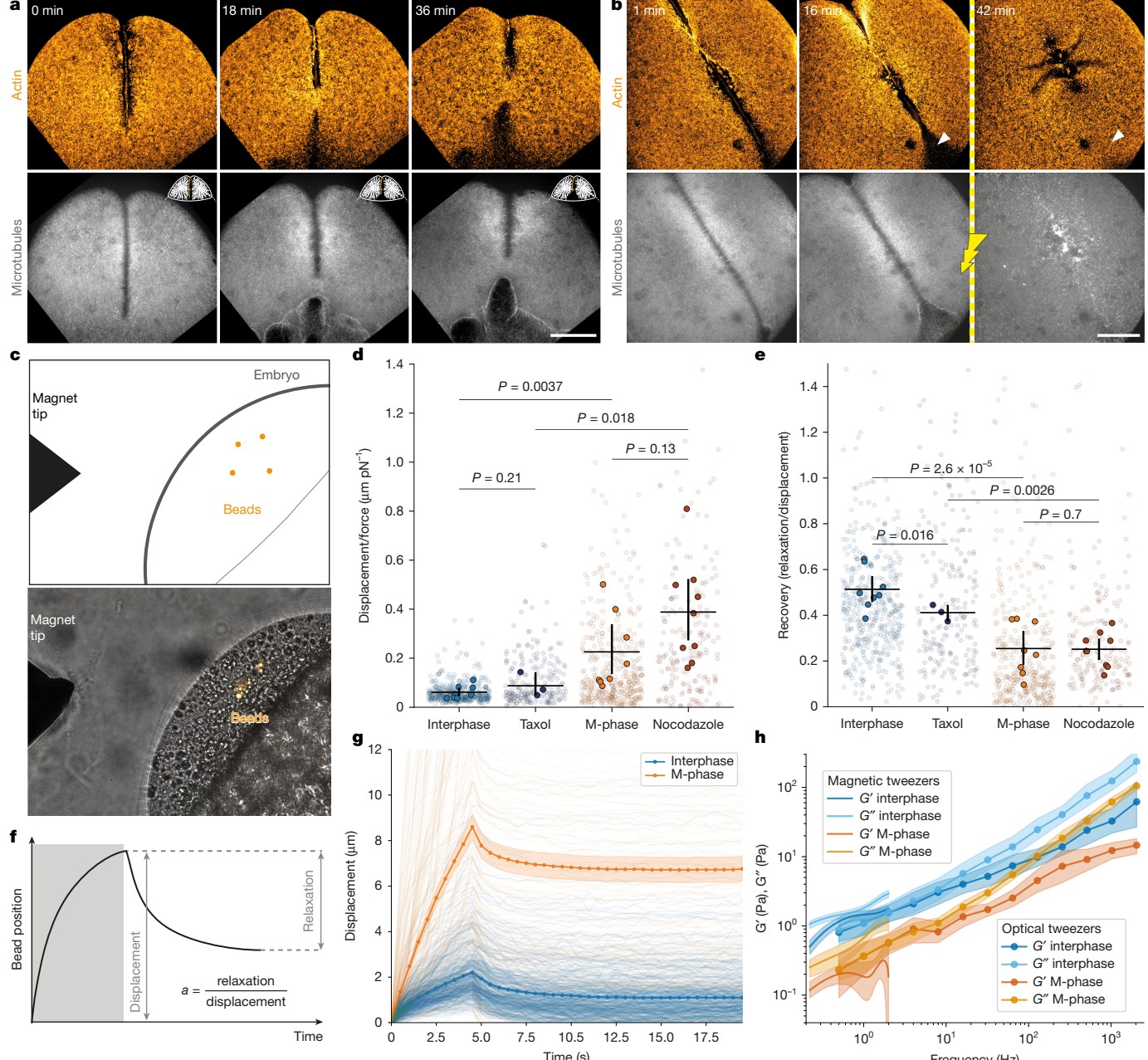

**Fig. 3 | Interphase microtubule asters drive cytoplasm stiffening. a**, Embryo (side view) treated with cycloheximide to arrested in interphase of the first cell division. Contractile band regression is visible in the middle image. Scale bar, 100 μm. **b**, Embryo treated with cycloheximide and SbTubA4P. The embryo is arrested in interphase, and microtubule depolymerization was activated after band regression onset (indicated by yellow line). White arrowheads indicate the area that ingresses and reverts to the previous state after photoactivation. Scale bar, 100 μm. **c**, Magnetic tweezers schematic (top) and brightfield image (bottom). The magnetic tip and embryo are in the field of view, the magnetic beads are shown in orange. **d**, Displacement of magnetic beads normalized to magnetic force in interphase ($n = 8$ embryos) and M-phase ($n = 8$ embryos; $P = 0.0037$), and with nocodazole ($n = 9$ embryos) and taxol ($n = 3$ embryos). Black lines represent bootstrapped mean and 95% confidence interval (1,000 resamples). Weighted two-sided Student's $t$-test with unequal variance.

**e**, Recovery (relaxation/displacement) of magnetic beads in interphase, M-phase, nocodazole-treated and taxol-treated embryos (as in **d**). **f**, Schematic of one magnetic pulse: the bead is displaced when the magnet is on and relaxes when the magnet is turned off. The grey shaded region indicates magnetic force application. Recovery of the beads is calculated as the relaxation over the displacement. **g**, Displacements of beads during magnetic pulses in interphase ($n = 430$ beads from 8 embryos) and M-phase ($n = 236$ from 8 embryos). Bootstrapped mean (solid line) and 95% confidence intervals of mean (shaded area, 1,000 resamples) for force ranges 20–60 pN. **h**, Bootstrapped elastic ($G'$) and viscous ($G''$) moduli obtained from weighted magnetic tweezers measurements ($n = 8$ embryos for interphase, $n = 8$ embryos for M-phase) conversion and optical tweezers measurements ($n = 12$ for interphase, $n = 12$ for M-phase; solid line shows the mean and shaded area represents 95% confidence intervals; 1,000 resamples).

slowed down when the embryo reached interphase (Supplementary Fig. 5a,b). When the embryo was treated with nocodazole the droplet in the cytoplasm extended homogeneously under force, regardless of the cell cycle stage (Supplementary Fig. 5a). Finally, consistent with the

cytoplasm behaving as a stiff material, the embryo surface can wrinkle[37] when the actin band contracts (Supplementary Fig. 5c). These wrinkles locally relax when the actin band is cut using laser ablation further supporting the local mechanical anchoring of the band to a stiff substrate

(Supplementary Fig. 5c and Supplementary Video 9). In addition, before the cell membrane ingresses, the cytoplasm self-organizes and divides into compartments[38–40], leaving an area between the two future cells that is devoid of microtubule asters. Consistent with the presence of microtubules leading to the stiffening of the cytoplasm, we observed that beads in the narrow region between the two asters—where microtubules are absent—behaved similarly to beads in M-phase, showing that the region between the asters remains fluid, as has been previously proposed based on cytoplasm flow patterns[41] (Supplementary Fig. 5e and Supplementary Video 10). We conclude that the microtubule asters drive changes in cytoplasm material properties that stiffen interphase cytoplasm with respect to M-phase.

## Cytoplasm fluidization drives ingression

Our results show that during interphase the contractile band is anchored and stabilized through stiffening of the bulk cytoplasm which mechanically supports its continuous growth. However, during this phase there is no ingression during early stages of band formation, followed by only slow and slight ingression towards the second half of interphase (Fig. 4a), raising the question of how the band can ingress into the stiff cytoplasm to divide it. This observation prompted us to carefully characterize the ingression until complete division. Consistent with the stiff cytoplasm in interphase, we found that the cell division is not completed during the first cell cycle[20], as the ingression is too slow to fully traverse the cytoplasm within the short cell cycle period (15–20 min total[42]). However, together with the cytoplasm fluidization in the next M-phase of the second cell cycle, the ingression velocity of the first band increased from $2.71 \pm 0.22$ µm min$^{-1}$ to $9.88 \pm 0.26$ µm min$^{-1}$, so that most of the ingression takes place in M-phase (Fig. 4a). To test whether this increase in ingression velocity is due to changes in contractility, we cut the contractile band during both phases. The cuts were made within the band to ensure that the surrounding material remained the same (Fig. 4c,d and Supplementary Video 11). We found that the contractility of the band, measured by initial recoil velocity, shows no significant difference ($P = 0.63$) in interphase ($0.32 \pm 0.18$ µm s$^{-1}$) compared with M-phase ($0.35 \pm 0.27$ µm s$^{-1}$), suggesting that the change in contractility of the band could not cause the drastic change in ingression velocity that we observe. Together, our results suggest that the stiff cytoplasm in interphase supports the band to be set up, but the fluidization of the cytoplasm in the next M-phase is necessary to allow ingression.

## Role of cytoplasm rheology

Our data suggest that the stiff cytoplasm allows the band to be set up, but the fluidization of the cytoplasm in the next M-phase might be necessary for ingression. To test that fluidization of the cytoplasm is sufficient to drive ingression of embryo and contraction of the band, we used a biophysical model. To phenomenologically capture the material properties of the cytoplasm in terms of simple springs and dashpots, we fit the magnetic bead experiments to the Jeffreys model for viscoelasticity[43] (Supplementary Fig. 4). We then modelled the cell as a spherical viscoelastic Jeffrey's material. The band was modelled as a contractile line that spreads across the equator through growth, while applying contractile forces along its length with uniform tension[44]. We used parameters fitted from the magnetic tweezers experiments and simulated a single cell cycle with interphase lasting 15 min and M-phase lasting 10 min (Supplementary Video 12). During interphase, the band grows faster than it contracts owing to the stiff cytoplasm, and it increases in length over time, while causing a small ingression. However, during M-phase the cytoplasm becomes more fluid, resulting in a large ingression during this period and a shortening of the total band length. The speed of ingression during M-phase is found to be 4.4 times faster than in interphase, similar to the experimental values. Our model was able to recapitulate the same ingression pattern, solely

by changing the material properties parameters (Fig. 4a,e), suggesting that a change in the material properties of the cytoplasm alone is sufficient to regulate the observed ingression.

To experimentally verify that fluidization of the cytoplasm drives faster ingression, we devised an experiment where microtubules are absent in interphase while the band is still present. In this situation, we expected the band to ingress faster even though the cell is in interphase. We used the photoactivatable microtubule inhibitor to prematurely depolymerize the interphase microtubule asters (Fig. 4f and Supplementary Fig. 6b), ensuring that the band had already started to grow but that the embryo was still in interphase. Although the microtubule depolymerization eventually causes the band to collapse, we found that the ingression velocity rapidly increases (from 2.6 µm min$^{-1}$ before the inhibitor activation to 9.0 µm min$^{-1}$ post-activation; $n = 17$) before the furrow retracts and the band collapses (Supplementary Video 13). We repeated this experiment in interphase-arrested embryos, to confirm that the embryos were indeed still in interphase when the microtubules were depolymerized. Again, we found that the furrow ingression increased after the microtubule depolymerization (from 5.1 µm min$^{-1}$ to 10.2 µm min$^{-1}$; $n = 4$), before the band started to collapse and the furrow retracted (Supplementary Fig. 6c). In summary, the stiff cytoplasm in interphase allows the band to extend around the cytoplasm (blastodisc), since it is stabilized through microtubule asters. Slight ingression in interphase is possible potentially owing to a gap between the asters that remains fluid (Supplementary Fig. 5e), but most of the ingression is enabled by the fluidization of the cytoplasm in M-phase.

## A mechanical ratchet for cell division

We found that fluidization of the cytoplasm in M-phase facilitates band ingression. However, this fluidization also leads to reduced anchoring of the band, which raises the question of how the band remains stable in M-phase. To answer this question, we imaged the ends of the band and measured their position over time. These measurements showed that the band is indeed unstable, meaning it retracts (Fig. 4b and Supplementary Fig. 5f–h), and it shortens at a rate of $10.25 \pm 0.20$ µm min$^{-1}$ (Supplementary Fig. 5h) over 4 min (consistent with previous microtubule perturbations), presumably owing to band contraction and reduced anchoring. However, the band did not fully collapse during M-phase, and its retraction was rescued when the interphase microtubule asters reappeared in the next cell cycle. The microtubule asters from the next cell cycle caused the formation of a new cleavage furrow perpendicular to the first one, but the ingression of this first furrow was also re-stabilized when the microtubule asters reached the cell-boundary, causing the retraction to stop. During this second interphase, the ends of the first band grew longer again (Supplementary Fig. 5f, 30 min onwards), while new bands formed for the next division simultaneously. When the cell cycle progressed to M-phase, the ingression of the first furrow continued further. The alternation between stability and growth in interphase, versus band instability and shortening but ingression in M-phase, persisted across several cell cycles until division was completed (Fig. 4g and Supplementary Video 14). This process resembles a mechanical ratchet, driving cell division without necessitating a fully formed actomyosin ring.

## Summary and discussion

The vast cell size and yolk of most early embryos create a unique cell geometry in which a complete contractile ring does not form during cell division. Our results reveal how an incomplete contractile band that only partially surrounds the embryonic cell can remain stable, grow and ingress into the cytoplasm despite having loose ends. As the ends of the contractile band never meet to form a closed ring, a mechanism distinct from the traditional purse-string model is required to explain how

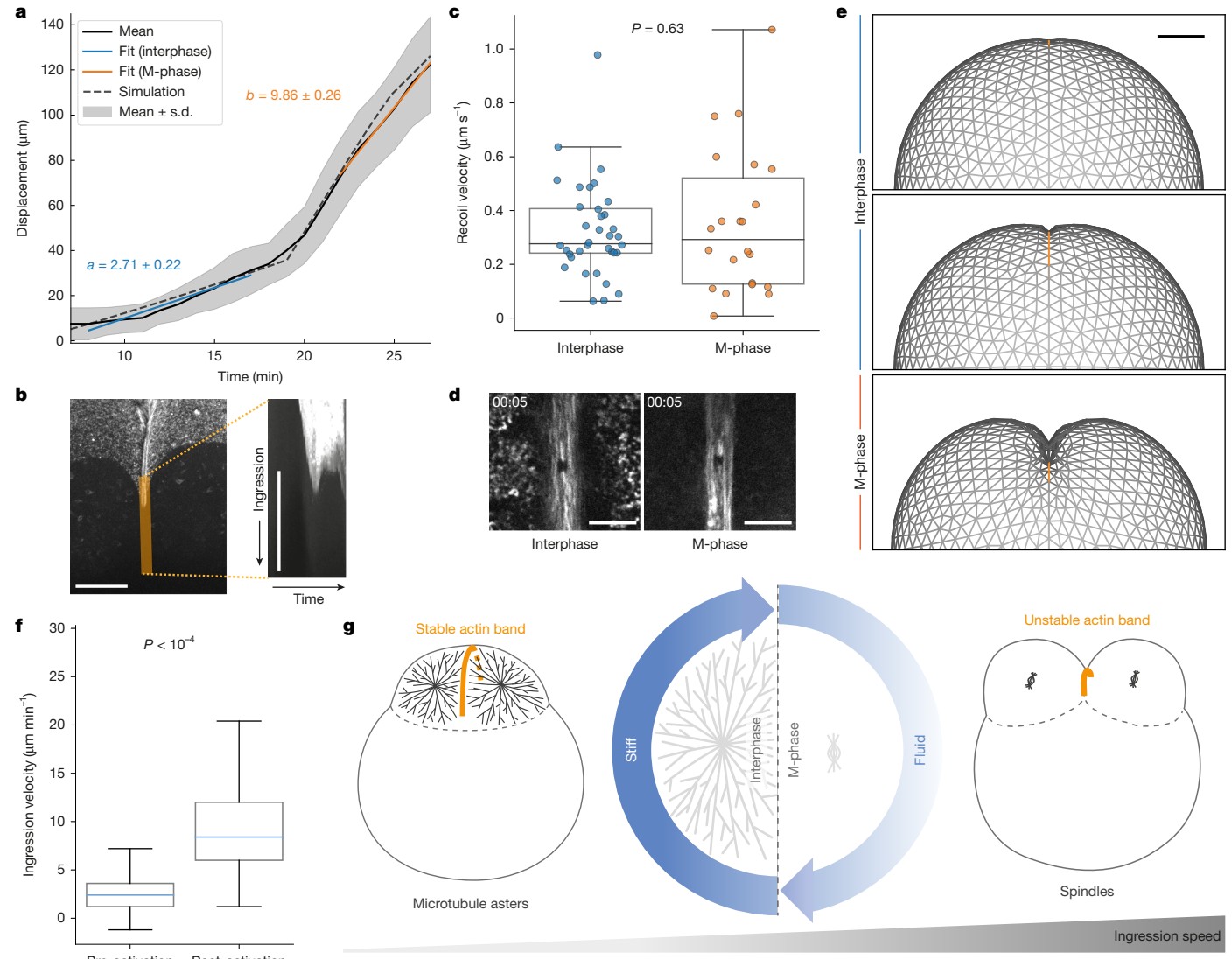

**Fig. 4 | A temporal ratchet mechanism prevents the band from collapsing.**
**a**, Ingression (displacement) of the contractile band, measured at the centre of the band, in interphase ($a = 2.71 \pm 0.22\ \mu m\ min^{-1}$) and in M-phase ($b = 9.86 \pm 0.26\ \mu m\ min^{-1}$) in $n = 6$ embryos (mean ± s.d.), and displacement of the contractile band in the simulation in more stiff ($2.43\ \mu m\ min^{-1}$) or fluid ($10.80\ \mu m\ min^{-1}$) surroundings. **b**, Ingressing contractile band in embryo. Actin is shown in grey (left) with a highlighted orange region, where the kymograph was recorded. Right, kymograph of the ingressing band over time. Scale bars, 50 μm. **c**, Measurement of band contractility via laser ablation, showing initial recoil velocity of cuts within the contractile band in interphase ($0.32 \pm 0.18\ \mu m\ s^{-1}$, $n = 22$ from 6 individual embryos) and M-phase ($0.35 \pm 0.27\ \mu m\ s^{-1}$, $n = 36$ from 9 individual embryos). $P$ value from two-sided Student's $t$-test. **d**, Visualization of laser cuts within the contractile band, related to Fig. 4c. Scale bars, 10 μm.

**e**, Visualization of computational model of springs (grey, surface plotted) and extending contractile band (orange); front view. Scale bar, 100 μm. **f**, Ingression velocity of the contractile band before ($2.642\ \mu m\ min^{-1}$) and after ($9.004\ \mu m\ min^{-1}$) photoactivation of the microtubule inhibitor SbTubA4P during interphase in $n = 17$ embryos during the first cell division ($P = 0.0000123$, paired-sample $t$-test). **g**, Temporal ratchet model for band stabilization and ingression. In interphase the cytoplasm is solid, and the band is stable and grows. In M-phase the cytoplasm is fluid, and the band can ingress, but is unstable. The cycle repeats rapidly (15–20 min cell cycles), to complete the division. In box plots (**c**,**f**), the centre line shows median, the box shows interquartile range (25th percentile to 75th percentile), whiskers extend to 1.5× interquartile range, and outliers are data beyond whiskers.

cytokinesis proceeds with an inherently unstable band structure. Our data support a model in which cell cycle-controlled stiffening of the cytoplasm drives unilateral cytokinesis via a temporal ratchet mechanism.

In line with other studies[32,33], we found that interphase microtubule asters induce changes in the material properties of the cytoplasm to become more stiff, probably through entanglement with endoplasmic reticulum and actin[45] and crosslinking through associated proteins[46]. We show that this stiffening has a key function by stabilizing the band, allowing it to form, but at the same time hindering its ingression. By contrast, fluidization of the cytoplasm during M-phase, when astral microtubules disassemble, facilitates the band ingression. The slight ingression we observed in interphase could be enabled by the fluid

region remaining between the asters. The otherwise lack of ingression during interphase due to stiffening of the cytoplasm is in line with the interphase arrested experiments, where the band does not fully penetrate the cytoplasm but only deforms it at the sides (Fig. 2e), as well as the taxol microtubule stabilization, where the band remains in place but does not ingress into the cytoplasm (Supplementary Video 7). Our microtubule depolymerization experiments (Fig. 4f and Supplementary Fig. 6b,c) and computational model corroborate these findings, indicating that changes in the material properties alone can account for changes in ingression velocity.

Our findings demonstrate that the actin band is anchored to microtubules. Although the molecular basis of this actin–microtubule

interaction remains unclear, the physical connection between actin and microtubules is apparent from our laser severing experiments, where microtubules splayed open when the actin band recoiled. Notably, this attachment is local and continues along the band, allowing local ingression even when the band is discontinuous, in contrast to models that depend on end-point anchoring or complete ring connectivity[47]. Intuitively, the band also must be anchored to the membrane, because the membrane ingresses following band ingression, indicating a connection between the two. However, the microtubule depolymerization experiments showed that neither the actin cortex nor the plasma membrane are sufficient to anchor the band stably, since the band collapsed following microtubule depolymerization.

Although cytoplasm fluidization allows faster ingression in M-phase, the contractile band becomes unstable and shortens. The rate of shortening, however, is slower than in the microtubule-depolymerized case, suggesting that there is some residual resistance, potentially provided by the furrow microtubule array[48]. Before the band collapses due to the shortening, the next interphase begins, and the first band is re-stabilized as the second division is initiated. This alternation between a solid cytoplasm state in interphase, allowing the band to be stable and grow, and a fluid state in M-phase, facilitating the band to ingress, repeats over several cell cycles until the division is complete. This balance results in a mechanical ratchet mechanism that ensures that unilateral cytokinesis proceeds despite the instability of an open-ended contractile band. Interestingly, the cortex also undergoes changes with respect to the cell cycle, and actin intensity in the cortex is reduced in a wave-like manner at the beginning of M-phase (observable in Supplementary Videos 5 and 14). This change could result in changes in cortex rigidity that could additionally facilitate the shape changes during ingression. Furthermore, excitable actin waves[49] can be observed in the cortex. These excitable waves are involved in furrow formation in other species[49], which highlights an additional instability in the (usually thought of as stable) furrowing process.

In the case of unilateral cytokinesis, the temporal ratchet mechanism offers an efficient solution to the challenges posed by the vast cell size and the rapid timing of the cell cycle. A fully formed contractile ring might take too long to grow and constrict around such large cells. Given that cytoplasmic division of early zebrafish embryos is inherently unstable, cell cycles must be rapid enough to prevent the development of such instability[38]. However, this short cell cycle-time appears to be insufficient to fully divide the cell. To ensure robust development with fast, synchronous divisions, the temporal ratchet provides a scalable mechanism for cell division that is employed during early embryogenesis. This mechanism persists in subsequent cell divisions until cells become separated from the yolk (starting from 16 cell stage), as well as small and symmetric enough to divide using a conventional contractile ring for division. We hypothesize that a complete ring would take longer to form and would therefore slow down the cell cycles, as in *Xenopus laevis*, where a complete ring forms during 30 min cell cycles[50], requiring twice as much time as the zebrafish embryo. In summary, we introduce a new model of cell division in which alternating material states of the cytoplasm enable ingression without a complete contractile ring.

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

## Methods

### Zebrafish lines and handling

Zebrafish (*D. rerio*) adults and embryos were maintained and handled according to established protocols[51].

The experiments were approved and licensed by the local animal ethics committee (Landesdirektion Sachsen, Germany; licence no. DD24.1-5131/394/33) and executed in accordance with the European Communities Council Directive 2010/63/EU on the protection of animals used for scientific purposes, as well as the German Animal Welfare Act. Sample sizes were chosen to allow statistically significant experiments while minimizing the number of animals used. Adult (4 months to 2 years of age) female and male zebrafish were used to produce embryos, and the sex of the embryos was not considered. Randomization and blinding were not applicable in this study. Transgenic animals had mixed background from AB and TL strains. Zebrafish transgenic lines used in this study are listed in Supplementary Table 1.

### Zebrafish sample preparation

Embryos were collected in E3 medium (5 mM NaCl, 0.17 mM KCl, 0.33 mM $CaCl_2$ and 0.33 mM $MgSO_4$) within 5 min after spawning and kept at 24–28 °C. Embryo clutch quality was inspected on a dissection stereomicroscope. Beads, fluorescent proteins, ferrofluid droplets or chemicals (SbTubA4P) were injected into the cytoplasm of one-cell stage transgenic zebrafish embryos according to[52]. Injection volumes were calibrated to 0.5 nl. Depending on the experiment, 0.5–1 nl were injected per embryo.

Embryos were mechanically dechorionated using forceps and mounted in 1% low-melting-point agarose (Sigma) in E3 medium supplemented with 25% w/v iodixanol (OptiPrep, STEMCELL Technologies 07820) for refractive index matching[53] on a CELLVIEW cell culture glass bottom dish (Greiner Bio-One 627860). Embryos were brought closer to the coverslip surface by keeping the dish upside down until agarose solidified, since the embryos float in the mounting media containing OptiPrep[54].

For magnetic tweezer, optical tweezer and band length experiments, dechorionated embryos were mounted in 0.5% low-melting-point agarose (Sigma) in E3 medium (without OptiPrep). Embryos were manually oriented before agarose solidification using a flamed glass capillary.

For measuring ingression velocity, embryos were mounted in 1% low-melting-point agarose (Sigma) in E3 medium (without OptiPrep) within their chorion. Embryos were manually oriented before agarose solidification using a flamed glass capillary.

### Chemical treatments

Dechorionated embryos at the one-cell stage were mounted in 0.5–1% low-melting-point agarose (Sigma-Aldrich), 25% OptiPrep Density Gradient Medium (OptiPrep, STEMCELL Technologies 07820) supplemented with 10 µg ml$^{-1}$ cytochalasin B from *Drechslera dematioidea* (Sigma-Aldrich), 200 µg ml$^{-1}$ cycloheximide (239763-1GM), 10 µg ml$^{-1}$ nocodazole (Thermo Scientific Chemicals 10762633) or DMSO control in a CELLVIEW cell culture glass bottom dish (Greiner Bio-One 627860).

### Light sheet microscopy

For the overview, data for Supplementary Fig. 1a were obtained using Zeiss Light Sheet Z.1, lateral illumination and detection geometry, equipped with 2 PCO edge 5.5 m, monochrome sCMOS cameras (SN 61000966, 61004862) for detection. Imaged with Zeiss Plan Apo 20× (1.0 NA) Water DIC objective. Controlled through Zeiss ZEN 2014 SP1 9.2.10.54 software.

### Confocal microscopy

For all confocal data except laser ablation, magnetic tweezers, and optical tweezers, the following microscope was used: Spinning disk confocal Andor Revolution platform with Borealis extension, Andor IX 83 inverted stand, Yokogawa CSU-W1 scan head, equipped with an Olympus silicone oil-immersion objective (30×/1.08 U Plan SApo, Silicone, OLYMPUS), or air objective (Olympus UplanXApo 20×/0.80 Air), recording with simultaneous imaging of two fluorophores with two Andor iXon Ultra 888, Monochrome EMCCD cameras (dexel size 13 µm).

For laser ablation, magnetic tweezers, and optical tweezers data, the following microscope was used: Spinning disk confocal Nikon Ti Eclipse, Yokogawa CSU-X1, equipped with a back-illuminated EMCCD camera (iXon DU-888 or DU-897, Andor) and a 60× water-immersion objective (Nikon Plan Apo VC 60× WI, NA 1.2) for laser ablation and optical tweezers, as well as 20× objective (Nikon Plan Apo 20× air, NA 0.75) for magnetic tweezers. Image acquisition was controlled by Andor iQ software in laser ablation and magnetic tweezer experiments, and through Nikon NIS-Elements software for optical tweezer experiments, as well as laser ablation shown in Fig. 2a.

### Epifluorescence microscopy

The data shown in Supplementary Fig. 5e were obtained with a Nikon Ti Eclipse microscope body using epifluorescence imaging through a 20× air, NA 0.5 objective, recorded with a Hamamatsu Flash 4.0 camera, controlled through µManager software[55].

### Laser ablation

For laser ablation, a mode-locked Ti:Sapphire laser (Coherent Chameleon Vision II) was coupled into the back port of the Nikon spinning disc microscope as previously described[54]. The custom-built laser ablation setup is based on a previously described layout[56]. Laser ablation was performed using a wavelength of 800 nm and typically a power of 1 mW after the pulse picker. Line cuts perpendicular to the contractile band were implemented by moving the sample with a high-precision piezo stage (PInano) relative to the stationary cutting laser. The cutting procedure was automatically executed by a custom-written software that controlled the piezo stage and a mechanical shutter in the optical path. The lengths of the cut were adjusted according to contractile band width (typically 20 µm cut length). To capture the fast dynamics of actin band recoil and recovery, single *z*-planes of the contractile band were acquired every 200–300 ms. The cut contractile band recovered at the ablation site within approximately 10 s and the subsequent cell divisions proceeded normally after ablation. For cuts within the band the length of the cut was 2 µm.

### Magnetic tweezers experiments

Fluorescent ferromagnetic beads were prepared by coating 2.8 µm Dynabeads Protein A (Invitrogen 10002D, Thermo Fisher Scientific) with anti-myc antibody, non-specifically labelled with DyLight 550 (Thermo Fisher Scientific 84530). The antibody-coated beads were additionally coated with polyethylene glygol as described[57]. Approximately 1–20 beads were injected per embryo into the cytoplasm at 1-cell stage. The samples were imaged on Nikon spinning disc confocal microscope (see above) for unperturbed embryos and on Nikon Ti Eclipse epifluorescence microscope for nocodazole and taxol-treated embryos.

Magnetic tweezers were home-built by inserting a pointed ferromagnetic rod into a solenoid. The rod was 6 mm in diameter and 120 mm long, made of HyMu-80 (Carpenter Technology). One end was sharpened to a 45° angle, which was introduced to the sample in the experiments. A 350-thread solenoid was made from a 0.5 mm diameter copper, wrapped around a Teflon support. Electric current was introduced to the solenoid with a voltage-controlled current generator, custom-built in house. Voltage, introduced to the current generator, was controlled with a laptop and a microcontroller (Arduino mini) though a custom script. The tip was mounted on an InjectMan NI2 micromanipulator (Eppendorf) for an easy and precise positioning of

the tip. The tweezers were mounted on a Nikon spinning disc confocal microscope or an epifluorescence microscope.

Forces produced by the magnetic tweezers were calibrated by displacing the 2.8 µm Dynabeads (same type of the beads as in the experiment) in glycerol at 20 C (viscosity of glycerol at 20 °C is $\eta$ = 1.412 Pa s). We ensured that the beads were at least 10 µm away from the glass and each other to avoid hydrostatic interactions with the surface. The voltage was set to 1,000 mV and the corresponding current through the solenoid was about 1,000 mA. The beads were tracked with TrackMate plugin in Fiji and the tracks were analysed with Python. Force–distance curves were obtained from 22 beads for the spinning disk setup and 59 beads for the epifluorescence setup (Supplementary Fig. 6a) and were fitted with a double exponential function, $f(x) = a_1 e^{-k_1 x} + a_2 e^{-k_2 x}$, which was used to infer the force on the beads in zebrafish.

### Optical tweezers setup

For optical tweezer measurements, Spherotech SPHERO Carboxyl Fluorescent Particles (CFP-2058-2), pink, 2.15 µm were coated with PLL-PEG as described[58] for passivation. Beads were injected into 1-cell stage zebrafish embryos as described above; bead solution was diluted to result in 5–20 beads per embryo.

The optical tweezer (Impetux, SENSOCELL) was set up as previously described[59,60]. In brief, the SENSOCELL setup was coupled to a Nikon Ti Eclipse confocal microscope by replacing the condenser with a calibration-free optical force sensor operating through detection of changes in light momentum. The 1,064 nm trapping laser was focussed onto the imaging plane of a 60× water-immersion objective (Nikon Plan Apo VC 60× WI, NA 1.2) creating an optical trap. The trap was manipulated using the manufacturers LightACE software.

### Active microrheology measurements with optical tweezers

The 1,064 nm trapping laser was used to trap a single bead and set to oscillate as described[60]. The oscillation frequencies used ranged from 0.5–6,250 Hz (data fitted up to 2,056 Hz), with a displacement amplitude of 1 µm and oscillation periods decreasing with increasing frequency (Supplementary Table 2). Measurements were made in interphase and M-phase (inferred by microtubule morphology) between 1-cell and 4-cell stage. The data were exported from the manufacturers RheoAnalysis software and further analysed using Wolfram Mathematica and Python (described 'Image processing').

### Photoactivation experiments

SbTubA4P was provided by O. Thorn-Seshold[30]. Prior to photocontrol experiments, the dose for inhibition was established using fully activated compounds as described[61]. Embryos were collected at 1-cell stage and injected with 1 nl of 1 mM – 2 mM SBTubA4P solution in the dark, only exposed to red light. Embryos were then mounted in agarose as described above and imaged on a confocal spinning disc microscope. The inhibitor was activated with 405 wavelength light either by taking a z-stack with a 405-laser or AMH fluorescent lamp, or by illumination of specific area using 405 nm Mosaic DMD.

For photoactivation experiments during interphase, where band ingression velocity was monitored, SbTubA4P was added to the mounting media, by diluting 2–3 µl of 10 mM SbTubA4P stock in 1 ml 0.5% low-melting-point agarose (at 37 °C) to a final concentration of 20–30 µM. Embryos were handled in the dark and the photoactivation was performed as above. For photoactivation in interphase arrested embryos, 10–20 µg ml⁻¹ cycloheximide were added to the mounting media in addition to SbTubA4P.

### Ferrofluid droplet and obstacle experiments

Ferrofluid was provided by O. Campas. Ferrofluid was injected into the cell of a 1-cell stage transgenic zebrafish embryo as described[36]. Magnet holder for magnetic field was 3D printed (material: PLA prusa Galaxy Black; printer: Prusa mini) based on a model obtained from

O. Campas and was equipped with grade N52 magnets (K&J Magnetics) as described[36]. The magnet holder was attached to the lid off CELLVIEW cell culture glass bottom dish (Greiner Bio-One 627860), to ensure reproducible spacing. To apply the magnetic field during imaging the lid and magnet holder were placed on the dish. To analyse the extension of ferrofluid droplets, z-stacks were turned into maximum intensity projections and an ellipse was fitted using the Fiji particle analysis plugin. The major axis of the ellipse was then plotted. Ferrofluid was found to nucleate microtubules and therefore used for obstacle experiments, by injecting approximately 1 nl into the centre of the embryo cytoplasm, without magnetic deformation.

### Image processing

All raw imaging data were processed in Fiji[62]. The raw tiff data were turned into maximum intensity projections.

**Laser ablation analysis.** For the laser ablation data in Fig. 1c, the microtubule channel was averaged across four time frames using the Fiji Walking Average tool. For the magnetic bead data shown in Supplementary Fig. 5e the bead channel was despeckled.

Recoil of the contractile band following laser ablation within the band was measured from kymographs. All the measured data were taken with 300 ms time interval and 0.12 µm pixel size. The data were divided into M-phase and interphase, based on microtubule morphology such as spindle presence, and in some cases DNA morphology such as nuclei or metaphase plate. Six embryos were measured in M-phase (11 measurements, recoiling ends plotted separately, therefore 22 points) and 9 embryos were measured in interphase (18 measurements, thus 36 recoiling ends). Lines with line width 7–9 pixels were drawn perpendicularly to the cut, and a kymograph was made from the line during the time from cut to end of recoil. The kymographs were cropped such that the time when the sample was still moving during the cut was removed, as this would influence the detected contours in later steps. The kymographs were processed in Python: the data were thresholded with the Otsu method and contours were detected using skimage.measure.find_contours[63]. The curves were fit with an exponential function $y = A(1 - e^{-\frac{x}{x_0}}) + c$, the initial recoil velocity $A/x_0$ was extracted and converted to µm s⁻¹. Only the first cut in any embryo was analysed. The P value was calculated using scipy.stats.ttest_ind (two-sided t-test) and the data were plotted in Python. In box plots, each whisker extends to the furthest data point in each wing that is within 1.5 times the interquartile range. Data points further than that distance are considered outliers and are marked with dots.

Recoil of the contractile band in comparison to the actin grey value of the band recovery were measured manually using the line tool in Fiji. A line was drawn within the cut between the two severed ends of the band. The line width was adjusted to be slightly below the width of the band. The line was adjusted at every time frame to match the length of the recoil, and the average grey value of the line was measured. For normalization, a line was drawn before the cut along the band as a normalization for the actin grey value. The recoil was normalized using the minimum and maximum value of the recoil length. Only the first cut in any embryo was analysed.

**Contractile band retraction analysis.** Contractile band retraction velocities following SbTubA4P or cycloheximide treatments were determined by processing the image with particle image velocimetry in Matlab (Matlab v.R2021a, PIVlab[64,65] v.3.02). For this analysis, nine embryos were measured for SbTubA4P and cycloheximide, respectively. PIV settings were: PIV algorithm: FFT window deformation, Pass 1: interrogation area 128 pixels, step 64 pixels. Pass 2: interrogation area 64 pixels, step 32 pixels. Sub-pixel estimator: Gauss 2 × 3-point (standard), Correlation robustness: Standard. Image pre-processing for PIV was: in Fiji: create input: maximum intensity projection, save as tif image sequence. In Matlab PIVlab: Enable CLAHE, Window size

64 pixels; auto contrast stretch. The PIV output data were further processed using Python. Each PIV dataset was aligned with the band along the $y$ axis. We took the average of the flow velocity within 25 μm of the $y$ axis, giving an average velocity $\bar{v}(y, t)$. From this, we determined the end points of the band as the points with the highest positive and negative speeds. This gave us a timeseries of the band end position over time and the flow speed at this point. From this, we calculated the average velocity of the band ends points, the average contractile flow speed at the end points, and the band growth speed as the difference between the two, both in M-phase and interphase. The $P$ value was calculated using scipy.stats.ttest_ind (two-sided $t$-test); error bars indicate the s.d.

Contractile band end retraction velocity during M-phase was measured using manual tracking plugin in Fiji[66] (v.2.1.1). The data for measuring was five embryos, imaged by mounting them on the side, so that the contractile band could be observed at its end (the one end that was facing the objective was measured per embryo). The embryos were imaged with a 20× air or 30× silicon oil objective on Andor spinning disk (see above), with 15–30 s time resolution. For analysis the data were averaged (Fiji Multi Kymograph Walking Average Plugin) over 2 timeframes for 30 s data and over 4 timeframes for 15 s data. The end of the band was plotted for five embryos together, normalized for the lowest value of each sample for alignment. The trajectories were fitted individually and the arithmetic mean of the slopes as well as the root mean square error were calculated to obtain the average retraction velocity.

**Band length measurements.** The total length of the contractile band was measured using the Simple Neurite Trace (SNT)[67], v.4.2.1 plugin in Fiji. The cursor auto-snapping and auto-tracing were disabled, and the band length was measured by manually tracing the band in the 3-dimensional $z$-stack in every time frame across 45 min (time interval 30–60 s) in 5 embryos. In one of the embryos, the entire band was measured, in the other four embryos, the centre of the band was defined as the centre from where it started to grow, and half of the band was measured (as the full band exceeded the image). The measurements of half of the band were multiplied by two in order to compare them to the measurement of the full band.

**Band ingression analysis.** Ingression of the band was measured using manual tracking plugin in Fiji[66] (v.2.1.1) in six embryos. The ingression was determined as difference between the lowest point of the ingression and the highest point of the cortex (imaged and tracked using actin (utrophin) or membrane (PH-Halo, kindly provided by P. Barahtjan, labelled with JF646)). To ensure that the ingression was not affected by mounting in agarose, the embryos were mounted within their chorion to avoid confinement. The cell cycle of the embryos was determined based on morphological criteria such as spindle or microtubule asters. The ingression velocity was determined by fitting a slope to the mean of the ingression from the six embryos in the respective cell cycle phases. The error represents the error of the slope.

**Microtubule orientation analysis.** Microtubule orientation analysis was performed manually as well as by automatic orientation detection. For manual orientation analysis, microtubules were traced using the Fiji line tool. The length, start point, end-point and angle was measured for every line. Seven embryos were analysed, with in total 712 measurements pre-cut and 720 measurements post-cut, covering over 100 randomly selected microtubules per image. In Python, all measured lines from pre- and post-cut embryos were plotted into the same plots, respectively, using slightly transparent lines colour-coded by angle.

For the automatic orientation analysis, the OrientationJ[68] plugin in Fiji was used. The parameters for microtubule orientation detection used were sigma: 5, Gradient: Gaussian, Grid size: 10, Length vector:

Maximum, Scale vector 80. Three embryos were analysed before ablation and 20 frames (6 s) after laser ablation. The mean orientations of the three embryos were plotted using Python and the lines were colour-coded according to their angle. For one embryo, the orientation profile over time was plotted (Fig. 2c) using Wolfram Mathematica.

**Magnetic tweezers analysis.** For magnetic tweezers experiments beads were tracked using the Fiji plugin Trackmate[69] v.7.12.1 (ref. 70). Trackmate parameters were stored in a table for every dataset, parameters were optimized to detect all beads in every time frame, to avoid gaps in tracks. False detections and detected beads outside region of interest were excluded. All parameters of the experiments (such as first magnet pulse, duration of pulse, duration of break, last pulse, presence of absence of microtubules, calibration of magnetic tip) were collected in a .csv file together with the file paths of the Trackmate data and further processed using Python[71,72,73].

First, the distance from the tip was calculated for each bead for each time point. The tip was detected manually in Fiji. The magnetic tweezer pulses were 5 s and the relaxation period before the pulse was 15 s. Due to natural reorganization of the cytoplasm, slow flows were produced on the beads. We corrected the bead displacement curves by subtracting this spontaneous flow. The flow was measured during the last 5 s of relaxation period, before the next force pulse was applied. As the relaxation times of the system were estimated to be around 1 s (less than 2 s in almost all cases (Supplementary Fig. 4f)), we assumed that the system has completely relaxed in 10 s. We measured the total displacement of each bead for each pulse and normalized it with the average force produced on the bead during the pulse. After the magnetic tip was turned off, the beads partially returned to their original position. We measured this relaxation distance and defined recovery $a$ as a ratio between the total displacement during the period with force applied and the relaxation during the period without external force.

We measured in total 827 individual tracks of beads across 12 embryos, of which 440 were measured in interphase and 387 were measured in M-phase. Number of tracks per embryo varies (Supplemenary Fig. 4l). As the magnetic force depends on the distance of the bead to the magnetic tip, the forces produced on beads were dispersed between 20 and 160 pN (Supplementary Fig. 4h,i). To compare bead responses between the phases across the same force range, we set the force range of interest to be between 20 pN and 60 pN, where most of the data for each phase lie. This leaves us with 674 full tracks (436 in interphase and 238 in M-phase) across 11 embryos (see Supplementary Fig. 4n for the distribution of tracks per embryo). Each bead went through on average 8 consecutive pulses and 17 at most, throughout which the measurements did not change significantly (Supplementary Fig. 4d,e).

We performed measurements in embryos treated with taxol and nocodazole, obtaining 211 tracks in 3 taxol-treated embryos and 154 tracks in 9 nocodazole-treated embryos in the 20 pN to 60 pN force range.

We analysed the normalized displacement and recovery $a$ for each bead separately, as well as averaged response per each embryo. We report normalized displacement and recovery of the averaged curve. For comparison between conditions, we perform statistical analysis on the averaged curves, calculating $P$ values with weighted unequal variance (Welch) $t$-test. Weights are calculated as the number of tracks per embryo, normalized with the total number of tracks and multiplied by the number of embryos, in each condition. We also report the weighted mean of the bootstrapped data ($n = 1,000$) and the 95% confidence interval.

Displacement curves were fitted with Jeffrey's model for viscoelastic material. Jeffrey's model consists of a Kelvin–Voigt element with a spring constant $k$ and a dashpot with viscous damping coefficient $\gamma_1$, in series with another dashpot with a viscous damping coefficient $\gamma_2$. This model predicts displacement of the bead $x(t)$ to be

$$x(t) = \begin{cases} \dfrac{F_0}{k}\left(1 - e^{-\frac{kt}{\gamma_1}}\right) + \dfrac{F_0 t}{\gamma_2}, & \text{for } t \le t_p, \\[2ex] x(t_p)\left(a e^{-\frac{kt}{\gamma_1}} - (1-a)\right), & \text{for } t > t_p, \end{cases}$$

where $F_0$ is an average force produced on the bead by the magnetic tweezers during the pulse of duration $t_p$. Recovery is calculated as

$$a = 1 - \frac{1}{1 + \frac{\gamma_2}{kt_p}(1 - e^{-kt_p/\gamma_1})}.$$

We performed statistical analysis (same as above) on the fit parameters obtained from the averaged curves in each phase. The mean values for each parameter were calculated with the weighted bootstrapping method ($n = 1{,}000$, see results in Supplementary Fig. 4f,g,j,k) and used for the theoretical prediction of the band ingression (described below).

**Optical tweezers analysis.** The averaged interphase and M-phase $G'$ and $G''$ measurements show a typical dependence of frequency consistent with a Kelvin–Voigt fractional[74], with functional form

$$G^*(\omega) = c_\alpha(i\omega)^\alpha + c_\beta(i\omega)^\beta$$

To compare M-phase and interphase material properties, we took the ratio of interphase to M-phase for $G'$ (and $G''$). This ratio is roughly constant for all frequencies (Supplementary Fig. 4b). To determine the fold change between interphase and metaphase, we bootstrapped the mean by taking the mean of randomly sampling with substitution from the total number of optical tweezers measurements in each condition, and then taking the mean of the ratio between conditions for low frequencies (between 0.5 and 16 Hz). We repeated this process $n = 1{,}000$ times to obtain a distribution of ratios, from which we obtain the mean fold change and 95% confidence intervals.

To further investigate the consistency between magnetic and optical tweezer measurements, we converted the magnetic tweezers creep compliance to $G'$ and $G''$ following Evans et al.[35]. In brief, we obtained the creep compliance from the averaged displacement curves (from eight embryos for each phase) of the magnetic tweezers experiments using $J(t) = 6\pi R\, x(t)/f(t)$, where $R$ is the bead radius, $x(t)$ and $f(t)$, the displacement and force applied to the bead respectively. Then, we applied the following expression to convert the creep compliance to $G^*$ (ref. 35):

$$\frac{i\omega}{G^*(\omega)} = i\omega J(0) + \frac{(1 - e^{(-i\omega t_1)})[J_1 - J(0)]}{t_1} + \frac{e^{-i\omega t_N}}{\eta}$$
$$+ \sum_{k=2}^{N} \left(\frac{J_k - J_{k-1}}{t_k - t_{k-1}}\right)(e^{-i\omega t_{k-1}} - e^{-i\omega t_k}),$$

where $J(0)$ is the compliance at $t = 0$, and $\eta$ is the steady state viscosity which is the extrapolation of the compliance at $t \to \infty$. To obtain $J(0)$ and $\eta$, we fit the magnetic tweezers curves to the phenomenological equation $b + a\,x + c e^{-dx}$, with $J(0) = b + c$, and $\eta = \frac{1}{a}$. We show the results for $G'$ and $G''$ of the weighted mean (bootstrapped $n = 1{,}000$) and the 95% confidence interval of the mean for 8 averaged creep responses for each phase (see section on magnetic tweezer image analysis).

## Theory

The cell is modelled as a spherical mesh of viscoelastic edges, of radius 350 μm, generated with roughly uniform density of vertices using Gmsh[75]. Notably, the mesh is symmetric across the $y$–$z$ plane, giving a line of springs around the equator, which will represent the band. Each edge is modelled by a Jeffrey's viscoelastic material, with parameters estimated from experiments for M-phase and interphase. The band was modelled by adding contractile tension to edges, starting from the top of the cell and going around the equator, with the ends of the band growing over time. We simulate the system starting with no band, with interphase lasting 15 min and M-phase lasting 10 min, with the material properties of the edges dependent on the current phase. We measure the ingression of the band by the displacement of the highest point of the band.

Each edge within the mesh is modelled as a viscoelastic Jeffrey's material, with parameters $k = 32.4$, $\gamma_1 = 56.1$ s, and $\gamma_2 = 189.8$ s in interphase, and $k = 21.4$, $\gamma_1 = 27.0$ s, and $\gamma_2 = 41.7$ s in M-phase, which were determined from the magnetic tweezers experiments. The constitutive equation for edge $i$ is given by

$$\sigma_i + \frac{\gamma_1 + \gamma_2}{k}\dot{\sigma}_i = \gamma_2 \dot{\epsilon}_i + \frac{\gamma_1 \gamma_2}{k}\ddot{\epsilon}_i$$

where $\sigma_i$ is the stress and $\epsilon_i = (l_i - l_{i,0})/l_{i,0}$ is the strain, with length $l_i$ and initial length $l_{i,0}$. Each edge may also be under a contractile tension $\lambda_i = \Lambda p_i$, where $\Lambda$ is the tension of the band, and $p_i$ is the proportion of the edge covered in the band ranging from 0 to 1. As we work with discrete elements, a band proportion between 0 and 1 represents an edge where the band only covers a fraction of it. Initially all edges have $p_i = 0$.

We simulate the band growth and cell mechanics numerically, with time step $dt = 0.01$ min. During each time step, we first grow the band and then move the vertices such that forces are balanced. The band growth is modelled by sequentially increasing the band amount $p_i$ of edges around the middle of the cell, starting from the top, until the total length of added band equals our desired amount, through the following procedure

1. Set the length of band to be added equal to $G = g\,dt$ where $g = 25$ μm min$^{-1}$ is the growth speed of the band, estimated from PIV data.
2. Starting from the centre of the band at $\theta = 0$, $\phi = 0$, and going around the edges of $\phi = 0$ as $\theta$ increases we attempt to add new band to the edge.
   a. If the band fully covers the band, $p = 1$, then we go to the next edge.
   b. If the amount of band to add is longer than the length available on the edge, $G > (1-p)l$, then we set the edge to be fully covered by the band, $p = 1$, and reduce the amount of band to be added by $G \to G - (1-p)l$.
   c. Otherwise, we added the remaining band to the edge, such that $p = p + G/l$ and finish growing the band.
3. Repeat step 2 for the $\phi = \pi$ direction, so that the band grows from both ends.

Next, we find the force balanced state. We discretize the constitutive equation as

$$\sigma_i(t_{n-1}) + \frac{\gamma_1 + \gamma_2}{k}\left(\frac{\sigma_i(t_n) - \sigma_i(t_{n-1})}{dt}\right)$$
$$= \gamma_2\left(\frac{\epsilon_i(t_n) - \epsilon_i(t_{n-1})}{dt}\right) + \frac{\gamma_1 \gamma_2}{k}\left(\frac{\epsilon_i(t_n) - 2\epsilon_i(t_{n-1}) + \epsilon_i(t_{n-2})}{dt^2}\right)$$

where $\sigma_i(t_n)$ and $\epsilon_i(t_n)$ are the stress and strain of edge $i$ at time $t_n = n\,dt$. The strain at time step $i$ is determined by the position of the edge's vertices, which are in turn determined by force balance. The force acting on vertex $j$ is

$$\boldsymbol{f}_j = \sum_i (\sigma_i + \lambda_i)\hat{\boldsymbol{t}}_i$$

where we sum over all edges connected to the vertex and $\hat{\boldsymbol{t}}_i$ is the unit vector along the edge pointing away from the vertex. The vertices are then moved in the direction of the force by a small amount, and new strains, stresses, and forces are calculated, and this process repeated until the force is approximately balanced, determined when the maximum force over all vertices is less than $10^{-5}$.

Finally, we have one free parameter to fit the experimental data with, the band tension $\Lambda$. To fit the experiments, we run simulations for a range of tensions, from 0 to 10, and select the tension with the lowest mean squared error between the predicted displacement in the model, and the mean measured displacement in experiments. We use a tension $\Lambda = 1.5$, which has a 99.4% $R^2$ score against experimental data.

## Statistics and reproducibility

Numbers of independent replicates of representative images (where not stated in figure legend).

- Fig. 1a: the band formation has been imaged in at least $n = 15$ individual embryos.
- Fig. 1b: the band formation at this higher magnification (40× or more) has been imaged in at least $n = 6$ individual embryos.
- Fig. 1d: the laser continuous laser ablation, over the time course of several minutes, has been repeated in at least $n = 3$ individual embryos, all demonstrating ingression of the remaining band that was not cut.
- Fig. 2a: The microtubule splay upon laser ablation has been imaged in at least $n = 10$ individual embryos.
- Fig. 2f: the band retraction upon microtubule depolymerization was repeated in at least $n = 10$ individual embryos.
- Fig. 2g: the band separation with the obstacle was repeated in at least $n = 2$ individual embryos.
- Fig. 3a: the band retraction in interphase arrested embryos experiment was repeated in at least $n = 10$ individual embryos.
- Fig. 3b: the band retraction in interphase arrested embryos followed by microtubule depolymerization was repeated in $n = 8$ embryos.
- Number of replicates of representative images in the Supplementary Figures.
- Supplementary Fig. 1a: at least $n = 10$ individual embryo replicates (with confocal spinning disc imaging instead of light sheet).
- Supplementary Fig. 1b: at least $n = 15$ individual embryo replicates
- Supplementary Fig. 1c: the band formation at this higher magnification (40× or more) has been imaged in at least $n = 6$ individual embryos.
- Supplementary Fig. 1d: the laser continuous laser ablation, over the time course of several minutes, has been repeated in at least $n = 3$ individual embryos, all demonstrating ingression of the remaining band that was not cut.
- Supplementary Fig. 1e: the band ingression without laser ablation has been imaged in at least $n = 6$ individual embryos at this magnification.
- Supplementary Fig. 2a: multiple subsequent cuts repeated in at least $n = 3$ embryos.
- Supplementary Fig. 2c: the cortex was cut in at least $n = 3$ individual embryo replicates.
- Supplementary Fig. 2d: the cortex and band were cut in at least $n = 3$ individual embryo replicates.
- Supplementary Fig. 2e: Actin inhibition was imaged in at least $n = 10$ individual embryo replicates.
- Supplementary Fig. 2f: The laser ablation upon wound-healing induction was imaged in at least $n = 1$ embryo.
- Supplementary Fig. 3a: the taxol treatment at the shown concentration was repeated in $n = 4$ individual embryos, from the same batch of embryos.
- Supplementary Fig. 3b: the taxol treatment at the shown concentration was repeated in $n = 4$ individual embryos, from the same batch of embryos.
- Supplementary Fig. 3c,d show different $z$-slices of the same embryo. The band retraction in interphase arrested embryos experiment was repeated in at least $n = 10$ individual embryos.
- Supplementary Fig. 5b: the ferrofluid droplet extension in untreated embryos was repeated at least $n = 13$ times in individual embryos.
- Supplementary Fig. 5c: laser ablation of the contractile band to resolve wrinkles was performed at least $n = 1$ time in individual embryos.
- Supplementary Fig. 5e: the demonstration of fluid region between asters was repeated at least $n = 2$ times in individual embryos.

## Reporting summary

Further information on research design is available in the Nature Portfolio Reporting Summary linked to this article.

## Data availability

The raw data are available at https://doi.org/10.25532/OPARA-947. Source data are provided with this paper.

## Code availability

The code for the magnetic tweezers analysis is available at https://github.com/ursaursic/cytokinesis_zebrafish/tree/for_paper. The code for the simulations of the band ingression is available at https://github.com/mstaddon/zebrafish_division. The code for plotting small datasets was uploaded with the respective raw data at https://doi.org/10.25532/OPARA-947.

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

**Acknowledgements** We thank O. Thorn-Seshold for sharing SbTubA4P and O. Campas for sharing Ferrofluid; the MPI-CBG Light Microscopy Facility; the MPI-CBG Fish Facility; the MPI-CBG Antibody facility; E. Pleško for assistance with the magnetic tweezers setup; M. Marass for providing editorial advice during the publication process; D. Oriola, O. Campas, M. Rinaldin, A. Honigmann, P. Tomancak, S. Diez and A. Mukherjee for scientific discussions; and D. Needleman, F. Decker, M. Loose and A. Lahee for comments on the manuscript.

J.B., A.K. and U.U. acknowledge support from the Deutsche Forschungsgemeinschaft (DFG, German Research Foundation) under Germany's Excellence Strategy–EXC–2068–390729961–Cluster of Excellence Physics of Life of TU Dresden. J.B. and U.U. acknowledge Volkswagen 'Life' grant number 96827.

**Author contributions** A.K. performed all experiments and performed data analysis except magnetic tweezers and optical tweezers. U.U. developed magnetic tweezers and performed magnetic tweezers experiments with A.K. and performed all magnetic tweezers data analysis. M.S. developed and performed numerical simulations and data analysis. J.B. and A.K. conceptualized the study. A.K. and J.B. wrote the manuscript with contributions from U.U. and M.S. J.B. supervised the work, performed optical tweezers data analysis and acquired funding.

**Funding** Open access funding provided by Max Planck Society.

**Additional information**
**Correspondence and requests for materials** should be addressed to Jan Brugués.

# Reporting Summary

## Statistics

For all statistical analyses, confirm that the following items are present in the figure legend, table legend, main text, or Methods section.

| n/a | Confirmed | |
|---|---|---|
| ☐ | ☒ | The exact sample size (*n*) for each experimental group/condition, given as a discrete number and unit of measurement |
| ☐ | ☒ | A statement on whether measurements were taken from distinct samples or whether the same sample was measured repeatedly |
| ☐ | ☒ | The statistical test(s) used AND whether they are one- or two-sided<br>*Only common tests should be described solely by name; describe more complex techniques in the Methods section.* |
| ☒ | ☐ | A description of all covariates tested |
| ☒ | ☐ | A description of any assumptions or corrections, such as tests of normality and adjustment for multiple comparisons |
| ☐ | ☒ | A full description of the statistical parameters including central tendency (e.g. means) or other basic estimates (e.g. regression coefficient) AND variation (e.g. standard deviation) or associated estimates of uncertainty (e.g. confidence intervals) |
| ☐ | ☒ | For null hypothesis testing, the test statistic (e.g. *F*, *t*, *r*) with confidence intervals, effect sizes, degrees of freedom and *P* value noted<br>*Give P values as exact values whenever suitable.* |
| ☒ | ☐ | For Bayesian analysis, information on the choice of priors and Markov chain Monte Carlo settings |
| ☒ | ☐ | For hierarchical and complex designs, identification of the appropriate level for tests and full reporting of outcomes |
| ☒ | ☐ | Estimates of effect sizes (e.g. Cohen's *d*, Pearson's *r*), indicating how they were calculated |

*Our web collection on statistics for biologists contains articles on many of the points above.*

## Software and code

Policy information about availability of computer code

| Data collection | Data for magnetic tweezers was collected with custom code written in vs code, available under the following link: https://github.com/ursaursic/magnetic_tweezers_brugueslab. |
|---|---|
| Data analysis | We have included a code availability statement where the entire code is accessible. |

For manuscripts utilizing custom algorithms or software that are central to the research but not yet described in published literature, software must be made available to editors and reviewers. We strongly encourage code deposition in a community repository (e.g. GitHub). See the Nature Portfolio guidelines for submitting code & software for further information.

## Data

Policy information about availability of data

All manuscripts must include a data availability statement. This statement should provide the following information, where applicable:
- Accession codes, unique identifiers, or web links for publicly available datasets
- A description of any restrictions on data availability
- For clinical datasets or third party data, please ensure that the statement adheres to our policy

We have included a data availability statement, with a doi to access all data.

## Research involving human participants, their data, or biological material

Policy information about studies with human participants or human data. See also policy information about sex, gender (identity/presentation), and sexual orientation and race, ethnicity and racism.

| | |
|---|---|
| Reporting on sex and gender | na |
| Reporting on race, ethnicity, or other socially relevant groupings | na |
| Population characteristics | na |
| Recruitment | na |
| Ethics oversight | na |

Note that full information on the approval of the study protocol must also be provided in the manuscript.

# Field-specific reporting

Please select the one below that is the best fit for your research. If you are not sure, read the appropriate sections before making your selection.

☒ Life sciences ☐ Behavioural & social sciences ☐ Ecological, evolutionary & environmental sciences

For a reference copy of the document with all sections, see nature.com/documents/nr-reporting-summary-flat.pdf

# Life sciences study design

All studies must disclose on these points even when the disclosure is negative.

| | |
|---|---|
| Sample size | The sample size was chosen in order to ensure that we have enough samples to be able to rule out that observations were random. This allowed us to perform a statistical test (t-test) in the case of band retraction in different treatments and tension on the band. |
| Data exclusions | no data exclusion |
| Replication | Experiments were repeated in multiple embryos from different parents and on different days to ensure reproducibility. In addition, mechanical measurements were performed with two different methods and indicated the same trend. |
| Randomization | Randomisation was not relevant for this study. |
| Blinding | The tracking of the beads and detection whether the cell cycle was interphase or M-phase, and the plotting of the tracked data, were done by two individual people to reduce bias. |

# Reporting for specific materials, systems and methods

We require information from authors about some types of materials, experimental systems and methods used in many studies. Here, indicate whether each material, system or method listed is relevant to your study. If you are not sure if a list item applies to your research, read the appropriate section before selecting a response.

## Materials & experimental systems

| n/a | Involved in the study |
|---|---|
| ☐ | ☒ Antibodies |
| ☒ | ☐ Eukaryotic cell lines |
| ☒ | ☐ Palaeontology and archaeology |
| ☐ | ☒ Animals and other organisms |
| ☒ | ☐ Clinical data |
| ☒ | ☐ Dual use research of concern |
| ☒ | ☐ Plants |

## Methods

| n/a | Involved in the study |
|---|---|
| ☒ | ☐ ChIP-seq |
| ☒ | ☐ Flow cytometry |
| ☒ | ☐ MRI-based neuroimaging |

## Antibodies

| | |
|---|---|
| Antibodies used | anti-myc (human) antibody received from MPI-CBG antibody facility,labelled with DyLight 550 (Thermo Fisher Scientific 84530), were |

| Antibodies used | used to visualise magnetic beads. |
|---|---|
| Validation | Validation was not relevant since we used non-specific antibodies solely for labelling beads. |

# Animals and other research organisms

Policy information about studies involving animals; ARRIVE guidelines recommended for reporting animal research, and Sex and Gender in Research

| Laboratory animals | Zebrafish (Danio rerio) of the following strains: Tg(actb2:EGFP-Hsa.DCX), Tg2(actb2:mCherry- Hsa.UTRN), and Tg(actb2:EGFP-Hsa.DCX, actb2:mCherry-Hsa.UTRN) |
|---|---|
| Wild animals | study does not involve wild animals |
| Reporting on sex | the sex of embryos was not considered |
| Field-collected samples | study does not involve samples collected from field |
| Ethics oversight | The experiments were approved and licensed by the local animal ethics committee (Landesdirektion Sachsen, Germany; license no. DD24.1-5131/ 394/ 33) and executed in accordance with the European Communities Council Directive 2010/63/EU on the protection of animals used for scientific purposes, as well as the German Animal Welfare Act. |

Note that full information on the approval of the study protocol must also be provided in the manuscript.

# Plants

| Seed stocks | na |
|---|---|
| Novel plant genotypes | na |
| Authentication | na |

