## [Peer Review File · Nature]

A mechanical ratchet drives unilateral cytokinesis

Corresponding Author: Professor Jan Brugues

Version 0:

Reviewer comments:

Referee #1

(Remarks to the Author)

Review of "A mechanical ratchet drives unilateral cytokinesis" by Kickuth et al.

The classical view of cytokinesis presents cell fission as occurring via the constriction of a continuous ring of cortical actin filaments (F-actin) and myosin-2 (the "cytokinetic apparatus") that completely encircles the cell at its midplane (ie the plane defined by the separating chromosomes) and closes inward symmetrically. Because this contractile apparatus is continuous, it is effectively anchored to itself. That this view is suspect has been known for decades, based on the simple observation that many cell types undergo "unilateral" cytokinesis, characterized by the formation of cytokinetic furrows that cut through the cell starting at one end of the cell and thus cannot be anchored to themselves (Rappaport, 1996. Cytokinesis in animal cells. Cambridge U. Press). Moreover, cells that normally divide symmetrically can be induced to undergo unilateral cytokinesis simply by displacing the mitotic spindle toward one side of the cell (PMID: 15998801). These observations argue that the contractile apparatus must be anchored at many points along its length to other structures in the cell since it can obviously do its job without being continuous. Indeed, the idea that the contractile ring is anchored at many points along its length is at least 50 years old (Schroeder, 1975. Dynamics of the contractile ring. Molecules and cell movement. Raven Press). However, there is a big difference between the idea of multiple anchors and the demonstration of such anchors.

In the current study, the authors report that a) the cytokinetic apparatus in early zebrafish embryos is indeed anchored along its length; b) the anchors are microtubules; c) microtubules control the viscosity of the cytoplasm in a cell cycle-dependent manner; d) c in interphase, microtubule asters increase the viscosity of the cytoplasm, which both retards the ingression of the cytokinetic apparatus and stabilizes it; e) in M-phase, microtubule asters shrink (due to microtubule depolymerization) and thus decrease the viscosity of the cytoplasm, which facilitates the ingression of the cytokinetic apparatus but also destabilizes it; f) these various behaviors collectively lead to a cell-cycle regulated ratchet such that the cytokinetic apparatus ingresses during M-phase, but then stalls during interphase but is held in place by the microtubules, and then during the next M-phase, the apparatus recommences ingression.

This is a lot of ground for one study to cover and the degree to which the authors are successful varies in different parts of the paper:

A. Anchoring of the cytokinetic apparatus along its length

This is the strongest part of the paper. Even with questions about some of the technical aspects of the work (see below) the impact of apparatus severing is clear: it recoils only a limited distance along its length and, moreover, other parts of the apparatus can continue contracting (ingressing) even when one part is severed. These results are similar to older, studies of wound-induced contractile rings in single cells (PMID: 11502762) and provide direct evidence of multiple anchors along the length of the cytokinetic apparatus.

B. Microtubules, but not F-actin, serve as anchors for the cytokinetic apparatus along its length.

This part of the paper is weaker. The strongest evidence provided for this claim is the reported reorganization of the microtubules near the wound site following laser ablation which the authors refer to as splaying. If it could indeed be shown that following severing, microtubules consistently and immediately undergo changes in organization consistent with attachment to the cytokinetic apparatus, this would directly demonstrate anchoring. However, the data presented for this point are limited. In Figure 2A, the reader is presented with three images: one pre-severing, a second 10s after severing, and a third, 15s after severing. Within these images, a single microtubule (or, more likely, a single microtubule bundle) is highlighted; this is reported to be an example of a microtubule that is "splaying". Splaying normally refers to linear elements splitting apart from each other but such behavior is not apparent in the images. Rather, the microtubule/microtubule bundle in question appears to be at an oblique angle to the long axis of the cytokinetic apparatus, which would be expected if the recoiling apparatus was anchored to the microtubule. However, it is apparent that the microtubule is already at an oblique angle to the apparatus prior to severing (see 0 s time point). And, in fact, many of the microtubules/microtubule bundles are at oblique angles relative to the cytokinetic apparatus prior to severing. What is really needed here are higher magnification movies that document the behavior of all of the microtubules/microtubule bundles near and distal to the site of severing with a careful quantification of the angles of the microtubules before and after severing. The magnification is important: as presented, it is very difficult to determine what is happening to most of the microtubules because the magnification is too low. Time resolution is also important: physical anchoring predicts that the microtubule angles should change at exactly the same time as the recoiling cytokinetic apparatus. In some of the experiments, the authors are imaging at 1s intervals (e.g. Video S1) which should be adequate. Longer intervals are less helpful as it makes it more difficult to track the behavior of specific microtubules.

The experiment with the uv-induced microtubule depolymerization does not address anchoring because, as the authors seem to be aware, this manipulation will also result in the termination of cytokinetic signaling (ie activation of Rho). The oil droplet experiment also does not address anchoring. It simply shows that the presence of microtubules is associated with the presence of the furrow which, again, is explained the need for microtubule to activate Rho.

Three other points need to be made here. First, it is impossible to evaluate the claim that the actin "melts away" on either side of the nascent cytokinetic apparatus simply because no movies showing this behavior at sufficient spatiotemporal resolution are provided. It would greatly help the authors case if they replaced figures 1A and 1B with much higher magnification images that allow the reader to actually see what is happening with the F-actin and microtubules within the furrow region itself. As it is, it looks like there are not any microtubules where the F-actin is concentrated. Second, while it is possible that the cortex is not under tension, as proposed by the authors, this would be surprising, given the number of previous studies indicating that it is (PMID: 19846787; PMID: 28530659). Perhaps this is just a special feature of zebrafish embryos, but a more likely explanation is that the assay being used is not sensitive enough to reveal low levels of tension. Third, surely the cytokinetic apparatus has to be anchored to the plasma membrane, otherwise the plasma membrane would not furrow when the apparatus contracts. Can the authors distinguish between multiple anchors to the plasma membrane versus multiple anchors to microtubules? If not, they should probably soften their conclusions about microtubules.

C. Microtubules control the viscosity of the cytoplasm in a cell cycle dependent fashion.

This part of the paper is also strong. The combination of the correlation of the presence/absence of asters over the course of the cell cycle with viscosity changes supplemented by the experimental manipulations make a strong case for this idea.

D. In interphase, microtubule asters increase the viscosity of the cytoplasm, which both retards the ingression of the cytokinetic apparatus and stabilizes it.

The second part of this claim is supported by the observation that elimination of bulk microtubules after the furrow has stalled during interphase results in dissolution of the cytokinetic apparatus. The first part makes sense, but to document it the authors would have to show that they can speed up furrow ingression by locally reducing microtubules in the path of the furrow. Unfortunately, it is doubtful this could be done without eliminating Rho activation.

E. In M-phase, microtubule asters shrink and thus decrease the viscosity of the cytoplasm, which facilitates the ingression of the cytokinetic apparatus but also destabilizes it.

The facilitation of cytokinetic apparatus ingression as a result of microtubule loss was not demonstrated (and, for the reasons set out above would be difficult to do) but the fact that the loss of microtubules results in the dissolution of the stalled cytokinetic apparatus is consistent with the idea that microtubule loss destabilizes the cytokinetic apparatus.

F. These various behaviors collectively lead to a cell-cycle regulated ratchet such that the cytokinetic apparatus ingresses during M-phase, but then stalls during interphase but is held in place by the microtubules, and then during the next M-phase, the apparatus recommences ingression.

This is, to the best of my knowledge, a completely novel idea. It would be of great interest even if the specific mechanisms are different than those proposed by the authors. The data supporting this idea are, however, tenuous. Part of the problem is that the behavior of the furrow reported by the authors is anomalous: in most cell types (including embryos such as *Xenopus*, which also furrow unilaterally) the furrow does not pause between cell cycles, but rather ingresses continuously until it has closed around the midbody. Thus, it is in the interest of the authors to convincingly document this behavior for zebrafish by providing low-mag z-views of furrow ingression while imaging F-actin with either microtubules or DNA so that the reader can observe the ingression, stalling and resumption of ingression as a function of cell cycle stage. It isn't that I doubt the authors, it's just that it is hard to evaluate this notion with the data provided.

A second feature of this idea that is somewhat confusing as presented is that cytokinesis doesn't normally happen in either the peak of m-phase or the peak of interphase, but at the boundary between the two. That is, at activation of the APC and the resultant fall of Cdk1 activity, the astral microtubules grow as the cell cortex becomes competent to support cytokinesis (PMID: 29738735). In most cell types, furrowing commences after the astral microtubules have completely regrown, which means that according to the authors, the cytoplasm would already be in the viscous state and thus should retard furrow ingression. The idea could be salvaged if zebrafish embryos are just different, a point which could be supported by the experiment suggested above. Alternatively, the idea could be salvaged if different regions of the cortex have markedly different microtubule densities. For example, if microtubules are relatively sparse in the path of the cytokinetic apparatus, but more dense on either side of it, this would presumably mean that viscosity differences could promote ingression. At one point, I thought this was what the authors were arguing. But after reading the statement that microtubules are "present throughout the furrow", and reflecting on the fact that furrowing is commencing at the point where the microtubules are in an interphase-like state, I became confused. The good news here is that this confusion could presumably be alleviated by a clearer explanation of what the authors think is going on. Better still would be a quantitative map of viscosity in the region of furrowing and on either side of it but that might be too much to ask.

(Remarks on code availability)

Referee #2

(Remarks to the Author)

In this manuscript, Kickuth and colleagues investigate the mechanics of cytokinesis in the asymmetric large zebrafish embryonic cell. This poses a challenge to the cell, because the canonical actin ring is in fact an actin band that cannot encircle the whole cell. They show that this band is anchored by and mechanically supported by an interphase microtubule network. They show that this network assembles and disassembles in a cell cycle dependent manner and by doing so, changes the mechanical properties of the cytoplasm accordingly, which allows successive stabilization and ingression of the actin band and ultimately successfully (if incomplete) cytokinesis. Overall, I found this manuscript very exciting and pleasant to read. It tackles the largely unanswered question of asymmetric cytokinesis. The provided data is convincing, and the writing is overall streamlined, although at times a bit allusive. My main concerns come from the lack of direct causal demonstrations between key parts of the manuscript, which if addressed would greatly strengthen the manuscript (see below). Provided these major points are addressed, I would support publication of this exciting paper in Nature.

Major

- In figure 2, the authors make the argument that keeping the microtubule network longer should prevent retraction of the band by arresting the embryos in interphase. This seems rather indirect. Can the authors also use a microtubule stabilizing drug eg taxol? This will allow to uncouple the microtubule effect to the cell cycle effect.
- The authors show convincingly that viscosity and cell cycle are coupled. They make the claim that this goes through microtubules. This should be shown, perhaps by coupling the rheologic measurements with microtubule perturbations (nocodazole, taxol, photoactivatable inhibitor). There is a mention of nocodazole in Figure S3b, but this is not explained.
- Along similar lines, it would greatly strengthen the mechanism to directly show that cytoplasm fluidization controls band ingression, although I acknowledge that this seems like a complicated experiment to do. Perhaps something along the lines of 10.1016/j.molcel.2024.06.024
- I did not understand at all the part on confinement and the membrane (lines 300-310 or so). It is not clear what the authors are looking at and how the videos answer these questions. What is "membrane attachment", what is the membrane attached to? I think this part should be clarified.
- Video S4 shows absolutely beautiful waves of actin. What are these? Are they linked to spindle microtubule depolymerization? I am not asking for experiments along those lines, but I think the authors could comment on this.
- In figure 2e: I cannot see the microtubules at the band, shouldn't they be present according to the model?
- The model proposes 2 regimes: interphase/mitosis. However, from the data in Figure 4a: it seems like there are in fact 3 regimes: 8-10 min, 10-20, 20 onwards. Could the authors comment on this?

Minor

- The microtubule terminology is at times confusing. What do the authors mean with "asters"? Is it just MTOCs? Or just microtubules? Sometimes they also use astral microtubules which in my opinion should be avoided unless talking about the mitotic spindle astral microtubules.
- I am not a fan of the color code chosen (orange/grey/brown). It is sometimes very difficult to see. For example, in figure 2a, I cannot distinguish the two types of polymers, in Figure 3b it is very hard to distinguish the curves/fits.
- Some of the figures only show a beginning and end time point eg figure 2b: the text mentions an effect 3 min after activation of the inhibitor, but the first image shown is 8 min. It is in the videos, but the main figures should show it.
- I find the part on the measurements of the mechanical properties of the cytoplasm more difficult to read than the rest. I suggest that the authors re-write this part of the text with a broad audience in mind.
- Are the labels inverted in Video S6?
- The author say that the contractility of the band is similar in interphase and M-phase but the recoil in Video S9 seems to be quite a bit more pronounced in one case. I think there is also something mislabeled here.

- Is the mechanism also at play in subsequent cell divisions?
- Line 328: would a ring take longer to constrict than a band? In most cells, even relatively large one, cytokinesis is an extremely fast process.

(Remarks on code availability)

Referee #3

(Remarks to the Author)

The manuscript "A mechanical ratchet drives unilateral cytokinesis" by Kickuth et al describes a new model to explain the still unresolved puzzle of cytokinesis initiated by a non-closed, and presumably non-anchored actin band. The question addressed is very important, as such structures are often found, but have yet not been understood. Although the subject is very important, and the experiments and the resulting new mechanisms are of high quality, the manuscript in its current form suffers from several weaknesses that should be addressed before publication in Nature. Unfortunately, the paper is quite difficult to understand, and the provided figures are often not presented in a very advantageous way. Furthermore, there are several points where the conclusions are not fully covered by the data, and require additional experiments or analysis. However, the findings are overall very interesting, the subject is very important, and if presented more convincingly the authors might indeed establish here a new mechanism of a key process. Hence, if the authors manage to overcome the critical points I am listing below, I think this manuscript can be a very important contribution that will end up in textbooks about development, and hence justifies publication in Nature.

Main critics:

- The data does not support the authors' claim of a sol-gel transition. Although I understand what the authors want to say, a sol-gel transition is defined as a transition where the dominating modulus switches from viscous to elastic modulus, typically because a polymer network becomes sufficiently crosslinked to be percolated. This means that the elastic modulus dominates over the viscous modulus. In the data presented, we actually see that the viscous modulus always remains dominant, hence the material remains a sol. In fact, the closest we get to a gel transition is during M-phase, where the two moduli become more similar. However, this is clearly against the intuition given by Figure 3b, and against the message the authors want to give, which is largely referring to the drastic change of viscosity. The reason, why the material remains predominately viscous is that the Jeffrey's model is intrinsically modeling a solution (so a sol, not a gel) for long timescales, as the dashpot is in series with the Maxwell model. I suggest being more careful about the usage of sol-to-gel transitions. Focusing on the drastic drop in γ_2 is the relevant part of the model, but this makes the system simply less viscous in M-phase, and not dominant elastic in the interphase.
- Another problem is regarding the potential changes in the contractility of the band. The experiment of cutting the band is very nice, but the conclusion that similar recoil velocity corresponds to similar contractility is based on the assumption that the viscosity of the band is constant. Given that the authors have measured drastic changes in cytoplasm viscosity, it is not clear that the viscosity of the band remains constant. This needs to be addressed more critically. Can the authors do experiments that show that the viscosity of the band remains constant?
- It would be very helpful to provide a molecular mechanism that can explain the drastic changes in the mechanical properties of the cytoplasm. Is there a change in crosslinks? Microtubules are typically not considered to play a crucial role for the mechanical properties of cells, so it would be important to find a reason for the observed changes.
- General rigor: I am missing statistical tests in basically all figures. I could not see if the observed differences are significant, or can be explained just by the variance of the measurements. Furthermore, I am worried that many statements are simply drawn from single images presented, and the quantification of the data remains too sparse. An example is Figure 1. It is very hard to understand what the authors claim, as in Fig a, we see not the formation of an actin band, but the absence of actin at least at 5 and 10 minutes. I do understand that this is due to the imaging, but I fear that the authors will not convince a reader that the actin band exists based on these images. Additionally, a clear quantification of the band and its growth is absent. This is also true for fig 1c and d. It is very hard to connect the statements in the text with the image shown at in the figure. I strongly advise the authors, to a) provide a sketch of the system studied in the first figure. b) use this sketch to provide a clear definition of the interphase and M-phase. c) to reuse a small version of this sketch to explain at the different manipulations, what the reader actually sees in the figures.
- Unfortunately, the SI figures were to a large extent simply not readable. Typically, this is a minor issue, but I could really not read many labels because the image quality is so low that the letters just disappeared, and also the datapoints are partially gone. This is honestly a bit sad, given that the authors submitted the paper to one of the most prestigious journals.
- The used naming of the different phases is confusing. Typically, the creation of the cytokinesis band is part of the M-phase, but here it is attributed to the Interphase. The reason is surely that in the early Zebrafish development, everything happens so quickly, that the phases follow the classical nomenclature. However, as the paper aims at a general scientific audience, it is important to define what the authors refer to as interphase, since their definition is different from the commonly used definition (interphase stops with breakdown of nuclear envelope!).
- To make the data comparable with other rheology, I suggest to divide γ and k by $6\pi R$, which will result in a loss and storage modulus in units of Pa s and Pa.
- It is disturbing that the authors remove experimental data that does not fit the model, and then state that the model is a good fit. Of course it is, if you remove all data that does not fit the Jeffrey's model. It would be important to understand if the removed data was simply to noisy (in which case it should be removed) or if it would better fit another model.
- The authors infer from the measurement Fig. 1e, where the recoil stopped before the band healed, that for example local anchoring is fixing the band. However, a simple explanation is that already a 10% recovered band provides sufficient mechanical resistance to stop the recoil. Can the authors provide additional evidence, for example by cutting again after the

recoil has stopped (at the low percentage of recovery (5-6 seconds after initial cut). This would show that the stop is not because of the small, but potential sufficient recovery of the band.

- I do not see that the oil droplets interrupt signaling in figure 2c, as MT are growing off from it. This data suggests in my opinion rather a steric interaction that suppresses locally the band formation.
- In the mechanical measurements, it would be important to report how many cycles of the pull and relax measurement were done, simply to understand how many independent measurements that 332 bead displacements correspond to.
- I like the mechanical model presented in Figure 4e, but details are missing. How was the interaction between the band and the mesh modelled? If rigid, how can this be since the microtubules, which are suggested to be the rigid anchors undergo dynamical remodeling.

Minor comments:

- Line 85: I am not sure if the word 'melts' is to be used here. I understand what the authors mean, but melting suggests a phase transition, but the authors seem to suggest simply a removal of actin. Better stay precise in the wording.
- There is a typo in line 238: ... is acts ...

(Remarks on code availability)

I am a bit confused by the relaxation fit function defined in the utils.py as

```
f = (1-a)*np.exp(-t/tau_r) + a
```

this model seems not consistent with the formula given in the SI.

```
x(t_p)(a exp(-kt/γ_1)-(1-a))
```

I did not crosscheck the code with simulated data myself, but I would like to ask the authors to take a look to ensure there is no inconsistency and explain, either in the code itself or the SI, where the inconsistency with paper comes from. I suspect that I overlooked something, but I think this might also happen to other interested readers. Please explain this difference.

Version 1:

Reviewer comments:

Referee #1

(Remarks to the Author)

The authors have thoroughly addressed my concerns and, in so doing, have significantly bolstered the case for publication of this exciting work.

(Remarks on code availability)

Referee #2

(Remarks to the Author)

The authors have done a good job addressing my concerns. One important concern remains: reviewer 1 pointed out that microtubule depolymerization leads to RhoA activation which could accelerate ingression; as far as I can tell, the authors have not really addressed this point which I think is very important.

The rest of my concerns are minor:

The order of figures should be fixed; in particular Figs 2 and S3 (example: Fig. 2 starts with 2a-c, then 2f, then 2e, then 2d, then 2g; Fig. S3 goes from S3c,d then S3a,b). It is hard to read.

It would be useful to have general labels over the microscopy images, e.g. "interphase" "m-phase" and showing the treatment e.g. "Nocodazole" "cycloheximide" to be able to compare images without always dropping down to the legend (which again complicates the paper for no good reason).

I still do not understand the part about confinement (paragraph starting on line 364, which arrives at the end but is about Fig. S1). I also do not think it is bringing much (and it is also probably lacking a reference) and I would suggest removing it.

Regarding microtubule "splaying": The quantifications and shorter time step videos asked by reviewer 1 in their first review are nice, but the images are not incredibly clear at first glance. Maybe an inset following a few individual microtubules would help?

In line 268 the authors state that the stuff cytoplasm support the band to be set up; is this shown?

Line 293, typo: "is" still present ("is" is missing).

Fig. S6c has a rogue double bar in the middle.

(Remarks on code availability)

Referee #3

(Remarks to the Author)

I am very pleased with the revisions of the paper, and they addressed all my concerns. I am particularly impressed by the consistency between the optical tweezer and the magnetic tweezer rheology data. Indeed, I have not seen yet such a nice comparison between these techniques in living cells, so this is a very nice result. Also, I was very happy to see the SI-Figures in good quality.

In my opinion, the paper is ready for publication, and thanks to the changes made by the authors it is now much more understandable.

I have identified some minor points that the authors should correct before acceptance, but I do not need to review the paper again.

Line 47: Maybe put an 'early' in front of Zebrafish, as the described process is only true for the initial divisions.

Line 99: There is a full stop missing after the “)”

Line 177: Consistency. Write Fig. capital?

Figure 3a: The length of the scale bar is missing

(Remarks on code availability)

Version 2:

Reviewer comments:

Referee #2

(Remarks to the Author)

The authors have addressed all of my comments; it is an exciting manuscript and I am excited to see it in press.

(Remarks on code availability)

Referee #1 (Remarks to the Author):

Review of "A mechanical ratchet drives unilateral cytokinesis" by Kickuth et al.

The classical view of cytokinesis presents cell fission as occurring via the constriction of a continuous ring of cortical actin filaments (F-actin) and myosin-2 (the "cytokinetic apparatus") that completely encircles the cell at its midplane (ie the plane defined by the separating chromosomes) and closes inward symmetrically. Because this contractile apparatus is continuous, it is effectively anchored to itself. That this view is suspect has been known for decades, based on the simple observation that many cell types undergo "unilateral" cytokinesis, characterized by the formation of cytokinetic furrows that cut through the cell starting at one end of the cell and thus cannot be anchored to themselves (Rappaport, 1996. Cytokinesis in animal cells. Cambridge U. Press). Moreover, cells that normally divide symmetrically can be induced to undergo unilateral cytokinesis simply by displacing the mitotic spindle toward one side of the cell (PMID: 15998801). These observations argue that the contractile apparatus must be anchored at many points along its length to other structures in the cell since it can obviously do its job without being continuous. Indeed, the idea that the contractile ring is anchored at many points along its length is at least 50 years old (Schroeder, 1975. Dynamics of the contractile ring. Molecules and cell movement. Raven Press). However, there is a big difference between the idea of multiple anchors and the demonstration of such anchors.

In the current study, the authors report that a) the cytokinetic apparatus in early zebrafish embryos is indeed anchored along its length; b) the anchors are microtubules; c) microtubules control the viscosity of the cytoplasm in a cell cycle-dependent manner; d) c in interphase, microtubule asters increase the viscosity of the cytoplasm, which both retards the ingression of the cytokinetic apparatus and stabilizes it; e) in M-phase, microtubule asters shrink (due to microtubule depolymerization) and thus decrease the viscosity of the cytoplasm, which facilitates the ingression of the cytokinetic apparatus but also destabilizes it; f) these various behaviors collectively lead to a cell-cycle regulated ratchet such that the cytokinetic apparatus ingresses during M-phase, but then stalls during interphase but is held in place by the microtubules, and then during the next M-phase, the apparatus recommences ingression.

This is a lot of ground for one study to cover and the degree to which the authors are successful varies in different parts of the paper:

A. Anchoring of the cytokinetic apparatus along its length

This is the strongest part of the paper. Even with questions about some of the technical aspects of the work (see below) the impact of apparatus severing is clear: it recoils only a limited distance along its length and, moreover, other parts of the apparatus can continue contracting (ingressing) even when one part is severed. These results are similar to older, studies of wound-induced contractile rings in single cells (PMID: 11502762) and provide direct evidence of multiple anchors along the length of the cytokinetic apparatus.

We thank the reviewer for the supporting words on the direct evidence of multiple anchors along the length of the actin band.

B. Microtubules, but not F-actin, serve as anchors for the cytokinetic apparatus along its length.

This part of the paper is weaker. The strongest evidence provided for this claim is the reported reorganization of the microtubules near the wound site following laser ablation which the authors refer to as splaying. If it could indeed be shown that following severing, microtubules consistently and immediately undergo changes in organization consistent with attachment to the cytokinetic apparatus, this would directly demonstrate anchoring. However, the data presented for this point are limited. In Figure 2A, the reader is presented with three images: one pre-severing, a second 10s after severing, and a third, 15s after severing. Within these images, a single microtubule (or, more likely, a single microtubule bundle) is highlighted; this is reported to be an example of a microtubule that is "splaying". Splaying normally refers to linear elements

splitting apart from each other but such behavior is not apparent in the images. Rather, the microtubule/microtubule bundle in question appears to be at an oblique angle to the long axis of the cytokinetic apparatus, which would be expected if the recoiling apparatus was anchored to the microtubule. However, it is apparent that the microtubule is already at an oblique angle to the apparatus prior to severing (see 0 s time point). And, in fact, many of the microtubules/microtubule bundles are at oblique angles relative to the cytokinetic apparatus prior to severing.

We thank the reviewer for pointing out the limitation of adding only three images to illustrate the splay. As the reviewer points out this is an important piece of evidence for the anchoring of the actin cable to microtubules. To address this limitation, we have now added other examples with improved imaging of microtubules during laser ablation in the 1-cell stage zebrafish embryo, imaged throughout the laser cutting process, and visualising the microtubule orientation prior to cutting and post-cutting. The examples shown now do not have oblique microtubules prior to cutting. Furthermore, we quantified the microtubule orientations across 7 embryos prior to cutting and post cutting, showing the temporal dependence of the splay of the microtubules after the laser cuts, and during the response.

What is really needed here are higher magnification movies that document the behavior of all of the microtubules/microtubule bundles near and distal to the site of severing with a careful quantification of the angles of the microtubules before and after severing. The magnification is important: as presented, it is very difficult to determine what is happening to most of the microtubules because the magnification is too low. Time resolution is also important: physical anchoring predicts that the microtubule angles should change at exactly the same time as the recoiling cytokinetic apparatus. In some of the experiments, the authors are imaging at 1s intervals (e.g. Video S1) which should be adequate. Longer intervals are less helpful as it makes it more difficult to track the behavior of specific microtubules.

We have added figures and movies of high-resolution microtubule imaging, ensuring that the microtubules can now be clearly seen. We have also measured the microtubule angles before and after the cut as a function of time. Before the cut, the angle of microtubules is perpendicular to the cut (angle = 0 by convention). After the cut, the splay of the microtubules quickly increases, and we now plot its temporal dependence as well as spatial profile from the distance of the cut (**Fig 2b,c**). All laser ablation data is taken at 300 ms time intervals. We have added supplementary videos with corresponding time stamps. We believe these data convincingly show the splaying of microtubules after the cut with higher spatial and temporal resolution, supporting the idea that microtubules are anchored to the actin band.

The experiment with the uv-induced microtubule depolymerization does not address anchoring because, as the authors seem to be aware, this manipulation will also result in the termination of cytokinetic signaling (ie activation of Rho). The oil droplet experiment also does not address anchoring. It simply shows that the presence of microtubules is associated with the presence of the furrow which, again, is explained the need for microtubule to activate Rho.

We acknowledge that signalling is needed to form and maintain the contractile band. However, we wanted to test if the presence of microtubules would stabilise the band even if it was artificially disconnected. The droplet experiment was designed to complement the laser ablation, and to locally disrupt part of the band as well as the microtubules, without perturbing them elsewhere. This experiment could not be done with the photoactivation or laser ablation. The droplet only locally disrupts the microtubules and the band, which can be clearly seen in **video S5** and **Fig 2g**.

The goal of this experiment was to disconnect the actin band without disrupting signalling in other parts of the cell. The experiment shows that, although the band is discontinuous, the open ends are stable and do not retract. This is only the case provided that there are microtubule asters present throughout the cytoplasm. When the cell cycle progresses to metaphase and the microtubule asters disappear, then these open ends retract (**video S5**). This experiment shows that the band can only be stable when the bulk cytoplasm is filled with microtubules, as the furrow-microtubules collapse with the band in M-phase.

Three other points need to be made here. First, it is impossible to evaluate the claim that the actin "melts away" on either side of the nascent cytokinetic apparatus simply because no movies showing this behavior at sufficient spatiotemporal resolution are provided. It would greatly help the authors case if they replaced figures 1A and 1B with much higher magnification images that allow the reader to actually see what is happening with the F-actin and microtubules within the furrow region itself.

We agree with the reviewer that it was hard to see the in the images without the dynamic information. We have added high resolution data of the actin band during its formation (**Fig 1b, video S1**), as well as quantifications of actin intensity across the cleavage plane (**Fig S1e**), which show reduced actin signal between the contractile band and the cortex. We have changed the wording for describing the actin gap.

As it is, it looks like there are not any microtubules where the F-actin is concentrated. Second, while it is possible that the cortex is not under tension, as proposed by the authors, this would be surprising, given the number of previous studies indicating that it is (PMID: 19846787; PMID: 28530659). Perhaps this is just a special feature of zebrafish embryos, but a more likely explanation is that the assay being used is not sensitive enough to reveal low levels of tension.

We have now added new data on the contractile band formation that shows microtubules in proximity of the contractile band, while the actin cortex is not connected. Furthermore, our laser ablation data shows a physical connection between the actin band and the microtubules, when the microtubules clearly respond to physical perturbations of the actin band. While we agree that there is a possibility that the laser ablation could only detect the response of the actin band and microtubules but not the tension of the cortex, when inducing wound healing, we can detect tension in the cortex generated as a response to the wound. However, it is not essential to know whether the cortex is under tension for the band ingression as it is disconnected from it, and we show that microtubules are the main contributor to physically stabilising the band (as our previous experiments show). We have accordingly changed our statements about the tension on the cortex.

Third, surely the cytokinetic apparatus has to be anchored to the plasma membrane, otherwise the plasma membrane would not furrow when the apparatus contracts. Can the authors distinguish between multiple anchors to the plasma membrane versus multiple anchors to microtubules? If not, they should probably soften their conclusions about microtubules.

We agree that the contractile band must also be anchored to the membrane, since the membrane ingresses together with the band. However, our microtubule depolymerisation experiment clearly shows that anchoring to the membrane is not sufficient to stabilise the band, because the band collapses in the absence of microtubules. Therefore, we infer that the main mechanical anchor is indeed provided by the microtubules and not the membrane.

C. Microtubules control the viscosity of the cytoplasm in a cell cycle dependent fashion.

This part of the paper is also strong. The combination of the correlation of the presence/absence of asters over the course of the cell cycle with viscosity changes supplemented by the experimental manipulations make a strong case for this idea.

D. In interphase, microtubule asters increase the viscosity of the cytoplasm, which both retards the ingression of the cytokinetic apparatus and stabilizes it.

The second part of this claim is supported by the observation that elimination of bulk microtubules after the furrow has stalled during interphase results in dissolution of the cytokinetic apparatus. The first part makes sense, but to document it the authors would have to show that they can speed up furrow ingression by

locally reducing microtubules in the path of the furrow. Unfortunately, it is doubtful this could be done without eliminating Rho activation.

We thank the reviewer for this very insightful suggestion. As the reviewer suggests, reducing microtubules without perturbing the contraction of the band should lead to increase of ingression in interphase and would be a key experiment to demonstrate the role of microtubule-modulated material properties in actin band ingression. As the reviewer points out, it is very challenging to do that without disrupting the band. However, we thought that upon microtubule-depolymerisation, we should observe a transient speed up of ingression before the band starts disassembling and collapsing. To this end, we imaged embryos in the right orientation with high temporal and spatial resolution. Strikingly, this data shows a beautiful speed up of the ingression velocity, followed by the expected collapse. The velocities that we observe after microtubule depolymerisation are very similar to the ingression velocity in M-phase. Finally, to verify that the embryos were indeed still in interphase when we depolymerised microtubules, we combined the depolymerisation experiment with interphase arrest, and we again found an increased ingression velocity upon microtubule depolymerisation. An example embryo and the data are shown in **video S13** and **Figs 4f, S6c**. We believe this new data addition strengthens the conclusion that the fluidisation of the cytoplasm, driven by the microtubule remodelling in M-phase, is causal for the increased ingression velocity in M-phase, and we thank the reviewer again for suggesting it.

E. In M-phase, microtubule asters shrink and thus decrease the viscosity of the cytoplasm, which facilitates the ingression of the cytokinetic apparatus but also destabilizes it.

The facilitation of cytokinetic apparatus ingression as a result of microtubule loss was not demonstrated (and, for the reasons set out above would be difficult to do) but the fact that the loss of microtubules results in the dissolution of the stalled cytokinetic apparatus is consistent with the idea that microtubule loss destabilizes the cytokinetic apparatus.

Thank you for this comment. To support our hypothesis, we have added data on premature microtubule depolymerisation in interphase, causing increased ingression speed (as described above). Remarkably, the band simultaneously starts to collapse, causing a complete regression of the furrow, after the increased ingression velocity (**Fig S6b**).

F. These various behaviors collectively lead to a cell-cycle regulated ratchet such that the cytokinetic apparatus ingresses during M-phase, but then stalls during interphase but is held in place by the microtubules, and then during the next M-phase, the apparatus recommences ingression.

This is, to the best of my knowledge, a completely novel idea. It would be of great interest even if the specific mechanisms are different than those proposed by the authors. The data supporting this idea are, however, tenuous. Part of the problem is that the behavior of the furrow reported by the authors is anomalous: in most cell types (including embryos such as *Xenopus*, which also furrow unilaterally) the furrow does not pause between cell cycles, but rather ingresses continuously until it has closed around the midbody.

We thank the reviewer for the appreciation of the novelty of our work. We would like to distinguish between two different types of unilateral cytokinesis here: meroblastic and holoblastic unilateral cytokinesis. The zebrafish embryo, as well as the majority of fish (teleosts), birds, reptiles, cephalopods and other examples mentioned in the introduction, divide meroblastically, in which case the cell is never fully divided and thus no midbody is formed.

The *Xenopus* embryo divides holoblastically, where the cytokinesis starts from one side only, with a cytokinetic band, however in this case the ends of the band eventually meet on the other side, and the cell is fully divided. In this case the cell eventually forms a ring, which can stabilise the cytokinetic apparatus. However, since it starts out as an incomplete ring, similar mechanisms may be in place to stabilise the band during its growth.

Unfortunately, it is very challenging to live image the details of actin band ingression in *Xenopus* embryos due to their opacity. We have made the distinction between meroblastic and holoblastic cytokinesis more clear in the text. Whether this mechanism also extends to holoblastic cytokinesis will be interesting to investigate in future studies.

Thus, it is in the interest of the authors to convincingly document this behavior for zebrafish by providing low-mag z-views of furrow ingression while imaging F-actin with either microtubules or DNA so that the reader can observe the ingression, stalling and resumption of ingression as a function of cell cycle stage. It isn't that I doubt the authors, it's just that it is hard to evaluate this notion with the data provided.

We thank the reviewer for this important suggestion to convincingly show the different stages of the mechanical ratchet. Supplementary **video 14** shows a low-mag video of the embryo during the ingression, stalling, and resumption of ingression, together with the microtubule signal as well as schematics to clarify the cell cycle stage. We have now updated this video to show the ratchet behaviour in once piece, as well as a step-by-step walkthrough. The Kymograph in **Fig 4b** shows the fast ingression in M-phase, followed by retraction, stalling, and continued ingression.

A second feature of this idea that is somewhat confusing as presented is that cytokinesis doesn't normally happen in either the peak of m-phase or the peak of interphase, but at the boundary between the two. That is, at activation of the APC and the resultant fall of Cdk1 activity, the astral microtubules grow as the cell cortex becomes competent to support cytokinesis (PMID: 29738735). In most cell types, furrowing commences after the astral microtubules have completely regrown, which means that according to the authors, the cytoplasm would already be in the viscous state and thus should retard furrow ingression. The idea could be salvaged if zebrafish embryos are just different, a point which could be supported by the experiment suggested above.

We agree with the reviewer that cytokinesis in zebrafish and many other embryos during cleavage stage is different, because the cell divisions happen much more regularly (every 15-20 min in zebrafish) than in somatic cells (order of days). Due to this extreme time constraint, there are no gap-phases during early development. The terms M-phase and interphase are used to describe the mitotic stage (characterised through a mitotic spindle) and the interphase where no spindle is present, a nuclear envelope is formed and the microtubule asters span the cytoplasm, respectively, but we agree that the nomenclature may be a bit confusing. Therefore, we have now added a better description of these stages, as well as references for the microtubule aster structure, in the main text.

Since the cell cycles are simply too short to allow full division during one cycle, the cytokinesis is not coupled to an individual phase but rather continues over multiple cell cycles to divide these very large embryonic cells. As mentioned above, we have updated **video S14** to demonstrate the furrow ingression occurring over multiple cell cycles.

Alternatively, the idea could be salvaged if different regions of the cortex have markedly different microtubule densities. For example, if microtubules are relatively sparse in the path of the cytokinetic apparatus, but more dense on either side of it, this would presumably mean that viscosity differences could promote ingression. At one point, I thought this was what the authors were arguing. But after reading the statement that microtubules are "present throughout the furrow", and reflecting on the fact that furrowing is commencing at the point where the microtubules are in an interphase-like state, I became confused. The good news here is that this confusion could presumably be alleviated by a clearer explanation of what the authors think is going on. Better still would be a quantitative map of viscosity in the region of furrowing and on either side of it but that might be too much to ask.

We acknowledge that this phrasing is confusing and have changed it to a better description of microtubule morphology. The microtubules are present along the entire length of the furrow on either side of it, and they are connected to the contractile band as demonstrated by the laser ablation experiments.

The thin region between the two microtubule asters indeed does remain fluid during interphase, as we demonstrated with magnetic tweezers (see also the striking difference between beads in this region and the region within the asters in **fig S5e** and **video S10**). However, our data suggests that this thin fluid region is not sufficient to allow fast ingression, but we hypothesise that this region could facilitate the slight ingression that we see during interphase.

Finally, since the embryonic cells are so big, not all parts of the cytoplasm are in the same cell cycle stage at the same time. The microtubule asters grow in waves radiating from the centre of the future cell, and they also disassemble in waves starting from the centre. This behaviour causes the asters to soften in the centre first, while the part of the aster close to the furrow remains rigid. Since this is a very dynamic process, a map of the viscosity is not feasible due to the variability. Generally, we measure “interphase” when microtubule asters span the cytoplasm, and M-phase, when the cytoplasm is devoid of microtubule asters. In M-phase, we do not consider measurements from beads that are stuck in the spindle.

Referee #2 (Remarks to the Author):

In this manuscript, Kickuth and colleagues investigate the mechanics of cytokinesis in the asymmetric large zebrafish embryonic cell. This poses a challenge to the cell, because the canonical actin ring is in fact an actin band that cannot encircle the whole cell. They show that this band is anchored by and mechanically supported by an interphase microtubule network. They show that this network assembles and disassembles in a cell cycle dependent manner and by doing so, changes the mechanical properties of the cytoplasm accordingly, which allows successive stabilization and ingression of the actin band and ultimately successfully (if incomplete) cytokinesis.

Overall, I found this manuscript very exciting and pleasant to read. It tackles the largely unanswered question of asymmetric cytokinesis. The provided data is convincing, and the writing is overall streamlined, although at times a bit allusive. My main concerns come from the lack of direct causal demonstrations between key parts of the manuscript, which if addressed would greatly strengthen the manuscript (see below). Provided these major points are addressed, I would support publication of this exciting paper in Nature.

Major

- In figure 2, the authors make the argument that keeping the microtubule network longer should prevent retraction of the band by arresting the embryos in interphase. This seems rather indirect. Can the authors also use a microtubule stabilizing drug eg taxol? This will allow to uncouple the microtubule effect to the cell cycle effect.

We appreciate this comment and we understand the concern. To address it, we stabilised the microtubules using taxol. However, this perturbation causes less physiological microtubules structures compared to the microtubule asters that are stabilised and maintained by arresting the cell cycle (stabilising here meaning, the asters remain in place, rather than the microtubules themselves are stabilised).

The perturbation with taxol nevertheless directly stabilised the microtubules and caused the band to remain stable on the cortex and prevented it from ingressing, presumably because the cytoplasm was too stiff in taxol treated embryos (as measured by magnetic tweezers, see new **Fig 3d**). **Video S7** shows a taxol treated embryo for 45 minutes, in which the band remains stable but cannot ingress. It also shows an embryo in which the microtubule perturbation prevented the band from properly forming due to unphysiologically stabilised microtubules.

The authors show convincingly that viscosity and cell cycle are coupled. They make the claim that this goes through microtubules. This should be shown, perhaps by coupling the rheologic measurements with microtubule perturbations (nocodazole, taxol, photoactivatable inhibitor).

We thank the reviewer for this suggestion. We have added rheological measurements of the cytoplasm in embryos treated with nocodazole (causing complete microtubule depolymerisation), as well as taxol (stabilising microtubules beyond their regular state), to demonstrate that the change in viscosity that we observe is indeed caused by the presence of the microtubule asters. We show that, regardless of the cell cycle of treated embryos, nocodazole treatment is mechanically comparable to M-phase and taxol treatment to interphase.

There is a mention of nocodazole in Figure S3b, but this is not explained.

We agree with the reviewer that there was only a brief explanation of this nocodazole experiment in the main text. We have now improved this section of the text to clarify the effect of nocodazole on the cytoplasm material properties.

- Along similar lines, it would greatly strengthen the mechanism to directly show that cytoplasm fluidization controls band ingression, although I acknowledge that this seems like a complicated experiment to do. Perhaps something along the lines of 10.1016/j.molcel.2024.06.024

We thank the reviewer for this key suggestion, which in fact was also mentioned by another reviewer. As the reviewer points out, this is a complicated experiment to do. However, we are very excited to show that fluidisation of the cytoplasm in interphase by microtubule depolymerisation indeed caused increased ingress velocity. We thought that upon microtubule-depolymerisation, we should observe a transient speed up of ingress before the band starts disassembling and collapsing. To this end, we imaged embryos in the right orientation with high temporal and spatial resolution. Strikingly, this data shows a beautiful speed up of the ingress velocity, followed by the expected collapse. The velocities that we observe after microtubule depolymerisation are very similar to the ingress velocity in M-phase. Finally, to verify that the embryos were indeed still in interphase when we depolymerised microtubules, we combined the depolymerisation experiment with interphase arrest, and we again found an increased ingress velocity upon microtubule depolymerisation. An example embryo is shown in video S13. We believe this new data addition strengthens the conclusion that the fluidisation of the cytoplasm, driven by the microtubule remodelling in M-phase, is causal for the increased ingress velocity in M-phase, and we thank again the reviewer for suggesting it.

- I did not understand at all the part on confinement and the membrane (lines 300-310 or so). It is not clear what the authors are looking at and how the videos answer these questions. What is “membrane attachment”, what is the membrane attached to? I think this part should be clarified.

The membrane attachment refers to the two future cells that result from the first division. During the course of the division, the membranes of the two future cells attach to each other, which could add long term stability to the cleavage furrow. We have clarified this section of the paper.

- Video S4 shows absolutely beautiful waves of actin. What are these? Are they linked to spindle microtubule depolymerization? I am not asking for experiments along those lines, but I think the authors could comment on this.

There are three different phenomena in video S4 (now video S5) that could be interpreted as actin waves:

1. Excitable waves in the actin cortex. These waves are caused by activator-inhibitor coupling of Rho and actin and have been beautifully described by Bement et al. (Bement, William M., et al. "Activator–inhibitor coupling between Rho signalling and actin assembly makes the cell cortex an excitable medium." *Nature cell biology* 17.11 (2015): 1471-1483.)¹. We have commented on these waves and cited the corresponding references.
2. A wave of change in actin intensity at 17 – 20 min. This wave occurs every cell cycle when the embryo transitions to M-phase. We have not further investigated this wave, but we have added speculations about potential functions in the discussion.
3. Large scale “rings” across the entire embryo: these are an imaging artifact caused by the curvature of the embryo and rather large z-spacing. Therefore, in this maximum intensity projection, areas of the cortex between two z-slices appear darker.

We have commented on the waves described in point 1 and 2 in the discussion.

- In figure 2e: I cannot see the microtubules at the band, shouldn't they be present according to the model?

Indeed, the microtubules are present at the band and throughout the entire cytoplasm in this interphase arrested embryo, but we acknowledge that it is difficult to see the microtubules, because they fill the entire cytoplasm, resulting in a rather homogeneous signal in this maximum intensity projection. We have replaced the image for one with a better microtubule signal and have added a single z-slices of the embryo to better visualise the presence of microtubules (**Fig S3c, d**).

- The model proposes 2 regimes: interphase/mitosis. However, from the data in Figure 4a: it seems like there are in fact 3 regimes: 8-10 min, 10-20, 20 onwards. Could the authors comment on this?

We reason that the ingression initially is that slow, because the band is probably too short to ingress during early stages of band formation. We have commented on it in the main text.

Minor

- The microtubule terminology is at times confusing. What do the authors mean with “asters”? Is it just MTOCs? Or just microtubules? Sometimes they also use astral microtubules which in my opinion should be avoided unless talking about the mitotic spindle astral microtubules.

We have clarified the microtubule terminology and cited papers in which the term for microtubule asters in large embryonic cells is described.

- I am not a fan of the color code chosen (orange/grey/brown). It is sometimes very difficult to see. For example, in figure 2a, I cannot distinguish the two types of polymers, in Figure 3b it is very hard to distinguish the curves/fits.

We thank the reviewer for pointing out this issue. We have added single channel grey scale images of actin and microtubules in **Fig 2a**. We have updated Figure 3b (now 3g) to more clearly distinguish between the experimental data and the fits.

- Some of the figures only show a beginning and end time point eg figure 2b: the text mentions an effect 3 min after activation of the inhibitor, but the first image shown is 8 min. It is in the videos, but the main figures should show it.

We agree that the timing notation here was misleading. We have updated the timestamps so that the inhibitor activation is set to 0 min, so that it can be clearly understood how many minutes before/ after activation is shown in the images. We have also updated the subfigure to show the embryo at 3 minutes after photoactivation (now also labelled with 3 min). In addition, the full video of the experiment is shown in **video S4**.

- I find the part on the measurements of the mechanical properties of the cytoplasm more difficult to read than the rest. I suggest that the authors re-write this part of the text with a broad audience in mind.

We agree with the reviewer that this section was perhaps too technical. We re-wrote this part of the text entirely to be more streamlined and accessible to a broad audience.

- Are the labels inverted in Video S6?

We thank the reviewer for catching this error, we have now updated the labels.

- The author say that the contractility of the band is similar in interphase and M-phase but the recoil in Video S9 seems to be quite a bit more pronounced in one case. I think there is also something mislabeled here.

The videos shown are simply to visualise how and where the bands were cut. The quantification of all the cuts (**Fig 4c**) show that the recoil can be variable but that there is no significant difference between interphase and M-phase. We have updated the supplemental video with more comparable examples and ensured that the labelling is correct.

- Is the mechanism also at play in subsequent cell divisions?

Yes, the mechanism is also at play in subsequent divisions. However, since the cell size is halved with every division, the subsequent divisions may take fewer cycles to complete the division. We have added this point to the discussion.

- Line 328: would a ring take longer to constrict than a band? In most cells, even relatively large one, cytokinesis is an extremely fast process.

While this is not something we can test in the embryo, we hypothesise that a ring would take much longer to form around the whole embryo, thereby potentially making the cytokinesis slower. For example, in *Xenopus laevis* embryos, where the cytokinetic apparatus starts forming as a band, but continues to grow until it completely surrounds the cell, the cell cycles take 30 min (compared to 15 min in zebrafish). Presumably, once the entire ring is formed the total ingression process should be faster, since it can ingress from all sides. We have added this point to the discussion.

Referee #3 (Remarks to the Author):

The manuscript "A mechanical ratchet drives unilateral cytokinesis" by Kickuth et al describes a new model to explain the still unresolved puzzle of cytokinesis initiated by a non-closed, and presumably non-anchored actin band. The question addressed is very important, as such structures are often found, but have yet not been understood. Although the subject is very important, and the experiments and the resulting new mechanisms are of high quality, the manuscript in its current form suffers from several weaknesses that should be addressed before publication in Nature. Unfortunately, the paper is quite difficult to understand, and the provided figures are often not presented in a very advantageous way. Furthermore, there are several points where the conclusions are not fully covered by the data, and require additional experiments or analysis. However, the findings are overall very interesting, the subject is very important, and if presented more convincingly the authors might indeed establish here a new mechanism of a key process. Hence, if the authors manage to overcome the critical points I am listing below, I think this manuscript can be a very important contribution that will end up in textbooks about development, and hence justifies publication in Nature.

We thank the reviewer for the enthusiastic comments on our manuscript.

Main critics:

- The data does not support the authors' claim of a sol-gel transition. Although I understand what the authors want to say, a sol-gel transition is defined as a transition where the dominating modulus switches from viscous to elastic modulus, typically because a polymer network becomes sufficiently crosslinked to be percolated. This means that the elastic modulus dominates over the viscous modulus. In the data presented, we actually see that the viscous modulus always remains dominant, hence the material remains a sol. In fact, the closest we get to a gel transition is during M-phase, where the two moduli become more similar. However, this is clearly against the intuition given by Figure 3b, and against the message the authors want to give, which is largely referring to the drastic change of viscosity. The reason, why the material remains predominately viscous is that the Jeffrey's model is intrinsically modeling a solution (so a sol, not a gel) for long timescales, as the dashpot is in series with the Maxwell model. I suggest being more careful about the usage of sol-to-gel transitions. Focusing on the drastic drop in γ_2 is the relevant part of the model, but this makes the system simply less viscous in M-phase, and not dominant elastic in the interphase.

We very much appreciated this comment. As the reviewer points out, we were using "gel" as a looser term, and their comment has prompted us to do further analysis as well as new measurements of the material properties of the cytoplasm. In addition to more careful analysis of the magnetic tweezers data, we have added optical tweezer experiments to directly measure the cytoplasm rheology. We have also converted the creep response of the magnetic tweezer experiments into elastic and viscous moduli (G' , G'') to check for consistency between these two types of measurements. Remarkably, these two measurements are consistent in magnitude and trend, and both show a ~3-fold increase of both the viscous and elastic moduli that is consistent for all measured frequencies. Examining the optical tweezer data more closely, we see that for high frequencies the viscous modulus dominates over the elastic modulus. However, for low frequencies, these become comparable (with interphase elastic modulus slightly larger than the viscous one). This would suggest a viscoelastic solid, and the G' and G'' curves are consistent with a Kelvin-Voigt fractional (similar to what was reported before²). However, here we decided to refrain from making strong claims about the exact rheological model that best fits the data, as one can take the measured G' and G'' as good descriptors of the material properties per se. What is relevant (as the reviewer points out) is that we can now show that both elastic and viscous moduli are increase by 3-fold in interphase. For the magnetic tweezer creep response, several models could fit the data (Jeffreys or Kelvin-Voigt fractional for example), with different mechanical interpretations that could be extrapolated at temporal scales beyond the measured time. We believe that the conversion to G' and G'' together with the optical tweezers gives a more accurate picture of the rheology of the system. For the finite element model, we still use the magnetic tweezer fits to the Jeffreys model, as a simple phenomenological model that captures the response in terms of simple springs and dashpots.

Finally, we downplay the recovery after magnetic pulses, as these are very noisy and subject to active behaviours of the cell (such as cytoplasmic flows). The recovery response is difficult to interpret, as opposed to the displacement response, because in the latter we control the force applied.

In summary, instead of “gelation” we decided to use “stiffening”—where stiffness is defined as resistance to deformation upon external force application—which better reflects both an increase of viscosity and elasticity. We thank the reviewer again for pointing us in this direction, and we believe that the extension of the rheological measurements (including magnetic tweezers, optical tweezers, and magnetic droplets) provides a comprehensive description of the rheology of the cytoplasm in M-phase and interphase that convincingly shows stiffening during interphase. We are also excited to provide these complementary measurements that are typically performed separately^{2,3} within one system, and we think that providing evidence of their consistency will also be a valuable contribution to the field.

- Another problem is regarding the potential changes in the contractility of the band. The experiment of cutting the band is very nice, but the conclusion that similar recoil velocity corresponds to similar contractility is based on the assumption that the viscosity of the band is constant. Given that the authors have measured drastic changes in cytoplasm viscosity, it is not clear that the viscosity of the band remains constant. This needs to be addressed more critically. Can the authors do experiments that show that the viscosity of the band remains constant?

We thank the reviewer for this critical point. The drastic changes in material properties that we measure in the cytoplasm are driven by the microtubule cytoskeleton, as demonstrated in **Fig 3d and 3e**, where microtubule depolymerisation causes loss of the observed changes. The contractile band, however, is formed by actin, and it may well not follow the same changes in viscosity.

The important distinction here is between the interphase-contractility versus the M-phase-contractility. To address this question, we have added an experiment where we reduce the viscosity of the cytoplasm during interphase via microtubule depolymerisation, as also suggested by other reviewers. Whilst this perturbation causes the band to collapse, it shows that the ingress speed increases following the fluidisation, before the band collapses (**Fig 4f, S6c and video S13**). This finding shows that the interphase-contractility of the band is able to induce similar ingress speed when the cytoplasm material properties are changed, indicating that the band contractility is similar in both phases.

Direct measurements on the viscosity of the contractile band were unfortunately not possible, and we are unaware of any literature that shows otherwise. The band is too small and too two-dimensional to do rheology on it. The best experiment we could do was to show that the microtubule depolymerisation measurements are consistent with the band contractility remaining constant. While it is still true that the contractility of the band may increase, our computational model shows that the changes of material properties alone with a constant band tension are sufficient to predict the ingress we observe. In summary, the microtubule depolymerisation experiments and the simulations together suggest that the band tension remains constant during the two phases.

- It would be very helpful to provide a molecular mechanism that can explain the drastic changes in the mechanical properties of the cytoplasm. Is there a change in crosslinks? Microtubules are typically not considered to play a crucial role for the mechanical properties of cells, so it would be important to find a reason for the observed changes.

There are several possible molecular mechanisms that explain the stiffening of the cytoplasm. Microtubules are one of the main mechanical elements of the cell⁴. They are known to change the material properties of the cytoplasm, and have been found to be the dominant mechanical element during interphase². In addition, the presence of the microtubule asters causes colocalisation with actin and the endoplasmic reticulum (ER)⁵. This localisation of actin and the ER is not observed in the absence of microtubule asters. It is also known that microtubule associated proteins such as molecular motors and crosslinking proteins associate with microtubule asters⁶. Therefore, between M-phase (when the bulk cytoplasm is devoid of microtubules) and interphase (when microtubule asters span the cytoplasm) there are changes in crosslinks of the microtubules, since in M-phase the “base structure”—meaning the microtubule asters—does not exist to

be crosslinked or to recruit actin and the ER. We thus conclude that the drastic change in cytoplasm material properties can be explained by the presence of the microtubule asters, which contribute themselves, and by crosslinking proteins as well as by recruiting other cellular structures. The above information is included in the discussion.

- General rigor: I am missing statistical tests in basically all figures. I could not see if the observed differences are significant, or can be explained just by the variance of the measurements.

We apologise for this, and we have now added statistical tests to the quantifications.

Furthermore, I am worried that many statements are simply drawn from single images presented, and the quantification of the data remains too sparse. An example is Figure 1. It is very hard to understand what the authors claim, as in Fig a, we see not the formation of an actin band, but the absence of actin at least at 5 and 10 minutes.

We have added higher resolution data of the contractile band during its formation that clearly shows the presence of the band. What can be observed as absence of the band at this resolution is the area of the cortex surrounding the band, which is indeed devoid of actin signal. In addition to showing an example of higher resolution data, we have added a quantification of the actin signal across the band, that shows the lack of actin signal on either side of the band.

I do understand that this is due to the imaging, but I fear that the authors will not convince a reader that the actin band exists based on these images. Additionally, a clear quantification of the band and its growth is absent.

We have added better data (**Fig 1b, video S1**) and quantifications (**Fig S1e**) of the actin band during its formation.

This is also true for fig 1c and d. It is very hard to connect the statements in the text with the image shown at in the figure. I strongly advise the authors, to a) provide a sketch of the system studied in the first figure. b) use this sketch to provide a clear definition of the interphase and M-phase. c) to reuse a small version of this sketch to explain at the different manipulations, what the reader actually sees in the figures.

We have provided a sketch of the zebrafish embryo that shows actin and microtubules, in interphase and in M-phase, in Figure 1. We re-used the sketch to visualise the perturbations where appropriate.

- Unfortunately, the SI figures were to a large extent simply not readable. Typically, this is a minor issue, but I could really not read many labels because the image quality is so low that the letters just disappeared, and also the datapoints are partially gone. This is honestly a bit sad, given that the authors submitted the paper to one of the most prestigious journals.

We are sorry to hear this. In our version of the SI figures the quality is fine. We hope that this issue will be resolved for the resubmission. We will check this in the submission portal.

- The used naming of the different phases is confusing. Typically, the creation of the cytokinesis band is part of the M-phase, but here it is attributed to the Interphase. The reason is surely that in the early Zebrafish development, everything happens so quickly, that the phases follow the classical nomenclature. However, as the paper aims at a general scientific audience, it is important to define what the authors refer to as interphase, since their definition is different from the commonly used definition (interphase stops with breakdown of nuclear envelope!).

We agree that the atypical phases during early embryonic development can be confusing, so we have added a clearer description of the cell cycles phases in the text and have also included the two cell cycles phases in the schematic mentioned above.

- To make the data comparable with other rheology, I suggest to divide γ and k by $6\pi R$, which will result in a loss and storage modulus in units of Pa s and Pa.

We thank again the reviewer for this suggestion. As explained above, we have now converted the magnetic tweezer force-displacement curves into G' and G'' and directly compared them with the new optical tweezer data.

- It is disturbing that the authors remove experimental data that does not fit the model, and then state that the model is a good fit. Of course it is, if you remove all data that does not fit the Jeffrey's model. It would be important to understand if the removed data was simply too noisy (in which case it should be removed) or if it would better fit another model.

We agree with the reviewer that removing of the data should be done carefully. Initially, we removed the measurements that were either too noisy or significantly distorted due to spontaneous cytoplasmic flows. Now we consider all the full tracks and do not filter for anything. For the final data analysis and the fitting of the Jeffrey's model, to obtain the parameters, we use the averaged curves of tracks in each embryo in the 20 to 60 pN range.

- The authors infer from the measurement Fig. 1e, where the recoil stopped before the band healed, that for example local anchoring is fixing the band. However, a simple explanation is that already a 10% recovered band provides sufficient mechanical resistance to stop the recoil. Can the authors provide additional evidence, for example by cutting again after the recoil has stopped (at the low percentage of recovery (5-6 seconds after initial cut). This would show that the stop is not because of the small, but potential sufficient recovery of the band.

We agree that actin healing contributes to re-stabilising the band after laser ablation. Indeed, in the plot showing the recoil length and the actin signal, although the band is slowing down before actin builds up, the band stops completely after the actin band is healed. For the experiments, we cannot wait until the band has stopped recoiling because actin has already healed which would induce build up of further tension. This can be seen when cutting the band after enough healing, the band opens up again but to a lesser extent. However, if we continuously cut the band to try to prevent any healing of the band, we can observe that the recoil eventually stops. This experiment of course is difficult, and we cannot rule out that some actin was already healed, however, it looks like the recoil stops and it does not open again even when we keep cutting. Examples of two consecutive cuts as well as continuous cutting are shown in **Fig S2a**. We have softened our conclusions in the main text.

- I do not see that the oil droplets interrupt signaling in figure 2c, as MT are growing off from it. This data suggests in my opinion rather a steric interaction that suppresses locally the band formation.

We agree that the obstacle might also simply present a steric hindrance, which would similarly locally perturb band formation. We have updated the description of the experiment.

- In the mechanical measurements, it would be important to report how many cycles of the pull and relax measurement were done, simply to understand how many independent measurements that 332 bead displacements correspond to.

The measurements of material properties for interphase and M-phase were performed in 12 embryos, out of which 6 embryos were measured in both interphase and M-phase, 3 were measured in only interphase and 3 were measured in only in M-phase. The number of data points per embryo varies, which is accounted for in the statistical analysis with the weighted average and weighted Welch t-tests for obtaining p values. On average, each bead went through around 8 magnetic tweezer pulses, throughout which the measured

response did not change significantly. We have now added additional analysis and grouping to the SI (**Fig S4I-m**).

- I like the mechanical model presented in Figure 4e, but details are missing. How was the interaction between the band and the mesh modelled? If rigid, how can this be since the microtubules, which are suggested to be the rigid anchors undergo dynamical remodeling.

The band is modelled by the same viscoelastic springs as the cell, but it spreads around the cell and adds an increased tension to the edges across its length. We have included this information in the supplement.

Minor comments:

- Line 85: I am not sure if the word 'melts' is to be used here. I understand what the authors mean, but melting suggests a phase transition, but the authors seem to suggest simply a removal of actin. Better stay precise in the wording.

We have updated the description of the actin band formation and added additional data and quantifications to visualise what is happening. We have updated the wording of the actin description.

- There is a typo in line 238: ... is acts ...

We have corrected this typo.

Referee #3 (Remarks on code availability):

I am a bit confused by the relaxation fit function defined in the utils.py as

$$f = (1-a)*np.exp(-t/tau_r) + a$$

this model seems not consistent with the formula given in the SI.

$$x(t_p)(a \exp(-kt/\eta_1) - (1-a))$$

I did not crosscheck the code with simulated data myself, but I would like to ask the authors to take a look to ensure there is no inconsistency and explain, either in the code itself or the SI, where the inconsistency with paper comes from. I suspect that I overlooked something, but I think this might also happen to other interested readers. Please explain this difference.

We thank the reviewer for pointing to this potential inconsistency. We have double checked and rewritten the fit function for obtaining Jeffrey's fit parameters. It is now consistent with the SI text:

```
# rising
x_1 = F_0 / k * (1 - np.exp(-k * t / eta_1)) + F_0 * t / eta_2
# relaxing
x_2 = (F_0 / k * (1 - np.exp(-k * t_1 / eta_1)) + F_0 * t_1 / eta_2) * (a * np.exp(-(t-t_1) * k / eta_1) + (1-a))
```

References:

- 1 Bement, W. M. *et al.* Activator–inhibitor coupling between Rho signalling and actin assembly makes the cell cortex an excitable medium. *Nature cell biology* **17**, 1471-1483 (2015).
- 2 Hurst, S., Vos, B. E., Brandt, M. & Betz, T. Intracellular softening and increased viscoelastic fluidity during division. *Nature Physics* **17**, 1270-1276 (2021).
- 3 Najafi, J., Dmitrieff, S. & Minc, N. Size-and position-dependent cytoplasm viscoelasticity through hydrodynamic interactions with the cell surface. *Proceedings of the National Academy of Sciences* **120**, e2216839120 (2023).
- 4 Dogterom, M., Kerssemakers, J. W., Romet-Lemonne, G. & Janson, M. E. Force generation by dynamic microtubules. *Current opinion in cell biology* **17**, 67-74 (2005).
- 5 Pelletier, J. F., Field, C. M., Fürthauer, S., Sonnett, M. & Mitchison, T. J. Co-movement of astral microtubules, organelles and F-actin by dynein and actomyosin forces in frog egg cytoplasm. *Elife* **9**, e60047 (2020).
- 6 Hawkins, T., Mirigian, M., Yasar, M. S. & Ross, J. L. Mechanics of microtubules. *Journal of biomechanics* **43**, 23-30 (2010).

Referees' comments:

Referee #1 (Remarks to the Author):

The authors have thoroughly addressed my concerns and, in so doing, have significantly bolstered the case for publication of this exciting work.

We thank the reviewers for the positive response and the support.

Referee #2 (Remarks to the Author):

The authors have done a good job addressing my concerns. One important concern remains: reviewer 1 pointed out that microtubule depolymerization leads to RhoA activation which could accelerate ingression; as far as I can tell, the authors have not really addressed this point which I think is very important.

We thank the reviewer for this comment. We believe that there might be a misunderstanding on the role of microtubules in RhoA activation. Rho is expected to be depleted by microtubule depolymerisation (and not activated), since the microtubules cause Rho activation in first place (Bement et al.). Consistent with this paper, Reviewer 1 stated "...this manipulation will also result in the termination of cytokinetic signaling (ie activation of Rho).", referring to the need of microtubules to activate Rho, which is then more explicitly stated a sentence later "It simply shows that the presence of microtubules is associated with the presence of the furrow which, again, is explained the need for microtubule to activate Rho."

Our experimental evidence and computation simulations in line with constant band tension, even the eventual decrease of Rho activity does not prevent the increase of ingression velocity (it should decrease it) upon fluidisation of the cytoplasm. Reviewer 1, who raised the concern initially, considers that these experiments address all their concerns.

In summary, the increase of ingression velocity cannot be driven by decreased Rho activation.

Bement, W. M., Benink, H. A., & Von Dassow, G. (2005). A microtubule-dependent zone of active RhoA during cleavage plane specification. *The Journal of cell biology*, 170(1), 91-101.

The rest of my concerns are minor:

The order of figures should be fixed; in particular Figs 2 and S3 (example: Fig. 2 starts with

2a-c, then 2f, then 2e, then 2d, then 2g; Fig. S3 goes from S3c,d then S3a,b). It is hard to read.

We have updated the Figure labelling to better follow the text.

It would be useful to have general labels over the microscopy images, e.g. “interphase” “m-phase” and showing the treatment e.g. “Nocodazole” “cycloheximide” to be able to compare images without always dropping down to the legend (which again complicates the paper for no good reason).

We have added this information when space allowed.

I still do not understand the part about confinement (paragraph starting on line 364, which arrives at the end but is about Fig. S1). I also do not think it is bringing much (and it is also probably lacking a reference) and I would suggest removing it.

We have removed the paragraph to avoid confusion.

Regarding microtubule “splaying”: The quantifications and shorter time step videos asked by reviewer 1 in their first review are nice, but the images are not incredibly clear at first glance. Maybe an inset following a few individual microtubules would help?

We have not added an inset for space reasons, but we refer to Supplementary video 3, where 3 embryos are shown with high time resolution, in which the individual microtubules can be followed.

In line 268 the authors state that the stiff cytoplasm support the band to be set up; is this shown?

All the experiments we have performed demonstrate that the actin band can only be stable and grow in the presence of bulk microtubules (depolymerization by photactivation, obstacle, arrest and further depolymerization of microtubules). Our rheological data shows that the presence of bulk microtubules leads to a 3-fold stiffening of the cytoplasm with respect to M-phase. Fluidization of the cytoplasm by depolymerizing microtubules when the embryo undergoes the cell cycle, leads to the collapse of the actin cable and an increase of ingression speed. This suggests that the stiffening of the cytoplasm by bulk microtubules can mechanically support the cable as it grows under contractile tension and prevents its collapse. This conclusion is also supported by our model and the new perturbation where we fluidize the cytoplasm and observe both collapse and speed up of band ingression. Altogether we think that our suggestion that stiffening of the cytoplasm by bulk microtubules supports the band to be set up is backed up from our experiments.

Line 293, typo: “is” still present (“is” is missing).

We thank the reviewer for catching this error and we have corrected it.

Fig. S6c has a rogue double bar in the middle.

We confirm that this bar was not linked to the plot in any way and thus we have removed it.

Referee #3 (Remarks to the Author):

I am very pleased with the revisions of the paper, and they addressed all my concerns. I am particularly impressed by the consistency between the optical tweezer and the magnetic tweezer rheology data. Indeed, I have not seen yet such a nice comparison between these techniques in living cells, so this is a very nice result. Also, I was very happy to see the SI-Figures in good quality.

We thank the reviewer for the enthusiastic comments on our manuscript.

In my opinion, the paper is ready for publication, and thanks to the changes made by the authors it is now much more understandable.

I have identified some minor points that the authors should correct before acceptance, but I do not need to review the paper again.

Line 47: Maybe put an ‘early’ in front of Zebrafish, as the described process is only true for the initial divisions.

We have added ‘early’ for clarification.

Line 99: There is a full stop missing after the “)”

We thank the reviewer for catching this error and we have corrected it.

Line 177: Consistency. Write Fig. capital?

We corrected this.

Figure 3a: The length of the scale bar is missing

We thank the reviewer for catching this error and we have added the scale bar information.